# Structural and thermodynamic analyses of the β-to-α transformation in RfaH reveal principles of fold-switching proteins

**Philipp K Zuber[1†], Tina Daviter[2‡], Ramona Heißmann[1], Ulrike Persau[1], Kristian Schweimer[1], Stefan H Knauer[1]***

[1]Biochemistry IV – Biophysical Chemistry, University of Bayreuth, Bayreuth, Germany; [2]Birkbeck, University of London, Malet Street, Bloomsbury, London, United Kingdom

**Abstract** The two-domain protein RfaH, a paralog of the universally conserved NusG/Spt5 transcription factors, is regulated by autoinhibition coupled to the reversible conformational switch of its 60-residue C-terminal Kyrpides, Ouzounis, Woese (KOW) domain between an α-hairpin and a β-barrel. In contrast, NusG/Spt5-KOW domains only occur in the β-barrel state. To understand the principles underlying the drastic fold switch in RfaH, we elucidated the thermodynamic stability and the structural dynamics of two RfaH- and four NusG/Spt5-KOW domains by combining biophysical and structural biology methods. We find that the RfaH-KOW β-barrel is thermodynamically less stable than that of most NusG/Spt5-KOWs and we show that it is in equilibrium with a globally unfolded species, which, strikingly, contains two helical regions that prime the transition toward the α-hairpin. Our results suggest that transiently structured elements in the unfolded conformation might drive the global folding transition in metamorphic proteins in general.

**\*For correspondence:**
stefan.knauer@uni-bayreuth.de

**Present address:** †MRC Laboratory of Molecular Biology, Francis Crick Avenue, Cambridge Biomedical Campus, Cambridge, United Kingdom; ‡The Institute of Cancer Research, London, United Kingdom

**Competing interest:** The authors declare that no competing interests exist.

## Editor's evaluation

This fundamental and timely work provides insights into the structural basis and thermodynamics of fold-switching proteins involved in the antitermination of transcription. By comparing six fold-switching and single-folding KOW domains from different organisms the authors provide compelling evidence showing that fold-switching domains are less thermodynamically stable than their single-folding counterparts. Furthermore, the authors identify a second fold-switching member of the NusG superfamily, VcRfaH, and investigate the physical basis of this fold-switching transition. This work should be of great interest to scientists in the fields of protein folding (theory and experiment), structural biophysics, and advanced protein NMR spectroscopy.

## Introduction

Fundamental understanding of how proteins fold has ever been one of the most important questions in structural biology and it is still not answered, despite recent progress in protein structure prediction (*Jumper et al., 2021*; *Tunyasuvunakool et al., 2021*). Since the formulation of the 'thermodynamic hypothesis of protein folding' by Anfinsen (*Epstein et al., 1963*), it has been generally accepted that the amino acid sequence of a protein determines its three-dimensional structure and that a protein adopts only a single folded conformation, which is referred to as physiological state and which corresponds to its global energy minimum. This conformation, in turn, fulfills one distinct function. While this 'one sequence–one structure–one function' dogma holds true for most well-folded (globular) proteins, it has been challenged by several discoveries over the past decades. Among those are, for instance, (i) moonlighting proteins, which fulfill two completely unrelated functions (*Jeffery, 2014*;

*Jeffery, 1999*), (ii) intrinsically disordered proteins (IDPs), which do not adopt a defined secondary or tertiary structure at all, but sample an ensemble of sterically allowed conformations instead (*van der Lee et al., 2014*), and (iii), most strikingly, metamorphic proteins (also referred to as fold-switching proteins), which can reversibly interconvert between at least two well-defined conformations, sometimes in response to a molecular signal (*Murzin, 2008*).

The free energy landscape of globular, well-folded proteins is often portrayed as a rugged funnel, with the 'rim' corresponding to the multitude of random coil structures of the 'unfolded state' (U state) and the deepest point (global minimum in Gibbs free energy, *G*), representing the 'native' or 'physiological' state (N state). IDPs, in contrast, exhibit a rather flat energy landscape and no specific conformation is favored, that is, significantly populated. Fold-switching proteins are thought to reside in-between these two scenarios, i.e. their energy landscape may be funnel-like, but it shows at least two major minima, each representing a distinct, well-folded conformation. The various conformations of a fold-switching protein may differ in the following aspects: (i) the type of secondary structure (α-helices, β-strands, turns, random coil), (ii) the extent of secondary structure elements, and (iii) the tertiary structure, usually in combination with (i) and/or (ii) (*Dishman and Volkman, 2018*; *Kim and Porter, 2021*). Additionally, these states often exhibit different quaternary structures, for example, monomeric in one state versus multimeric in another state.

A particularly intriguing example of fold-switching proteins is the transcription factor RfaH from *Escherichia coli* (*Ec*RfaH), a member of the universally conserved family of NusG (bacteria)/Spt5 (archaea and eukaryotes) proteins (*Werner, 2012*). NusG/Spt5 proteins exhibit a modular structure

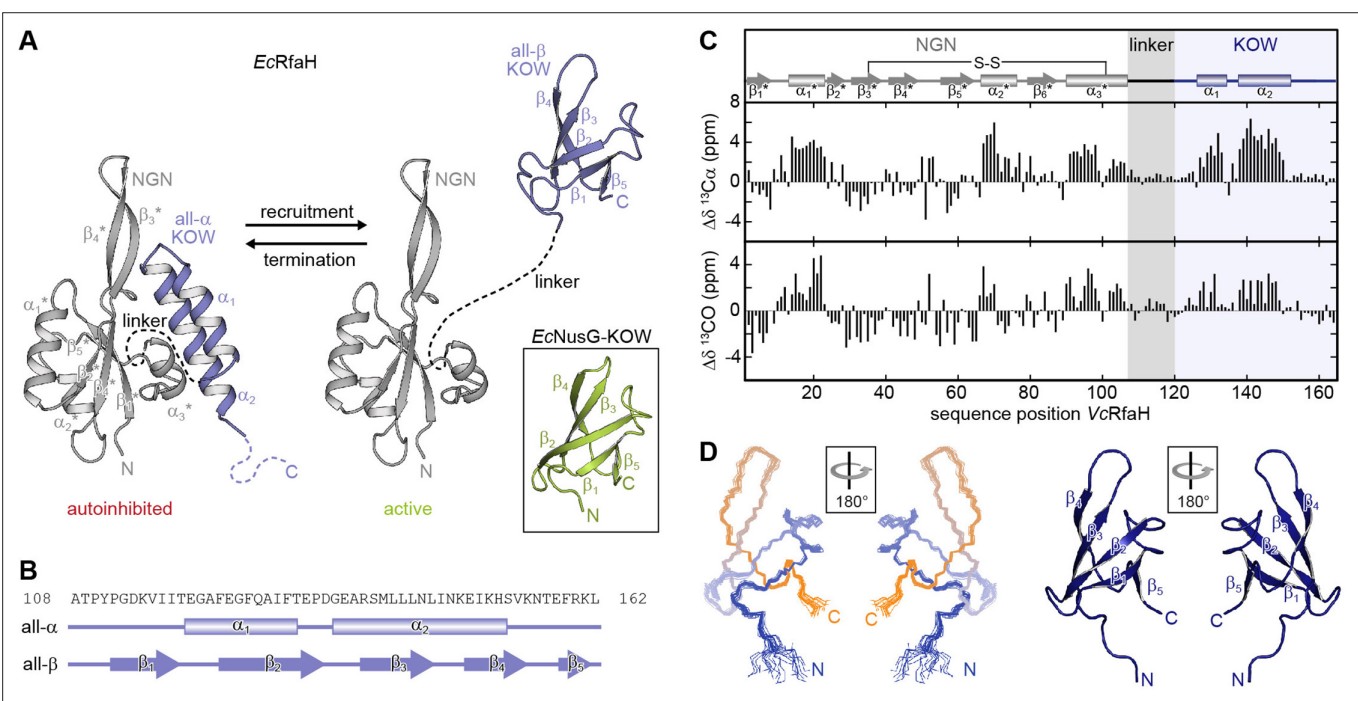

**Figure 1.** Fold-switching within the NusG/RfaH family. (**A**) Cartoon representation of *Ec*RfaH in the closed, autoinhibited state (left; protein data bank identifier (PDB-ID): 5OND) and in the open, active conformation (right; PDB-ID all-β *Ec*RfaH-KOW: 2LCL) as well as of *Ec*NusG-KOW (boxed; PDB-ID: 2JVV). Unstructured regions are shown as dashed lines, termini are labeled. (**B**) Secondary structures of *Ec*RfaH-KOW in the all-α and the all-β state. Tubes indicate α-helical elements, arrows represent β-strands. The amino acid sequence is shown above. (**C**) Secondary chemical shift of *Vc*RfaH. The plots show the difference between the observed chemical shift and the corresponding predicted random coil value of $^{13}$Cα (top) and $^{13}$CO (bottom). Positive values indicate helical, negative values elongated (β-sheet) structures, and values close to zero are observed for random coil-like structures. The secondary structure elements inferred from the analysis are shown above the graphs (code for secondary structure elements as in (**B**)). The position of the identified disulfide bridge (see also *Figure 1—figure supplement 1A, B*) is indicated. (**D**) Left: Ribbon representation of the 20 lowest energy structures of *Vc*RfaH-KOW (PDB-ID: 6TF4). Right: Cartoon representation of the lowest energy structure. β-Strands and termini are labeled.

The online version of this article includes the following figure supplement(s) for figure 1:

**Figure supplement 1.** Disulfide bridge formation in *Vc*RfaH.

**Figure supplement 2.** Structure comparison of Kyrpides, Ouzounis, Woese (KOW) domains used in this study.

with several domains. Bacterial NusG consists of at least an N-terminal domain and a C-terminal Kyrpides, Ouzounis, Woese (KOW) domain connected by a flexible linker (*Werner, 2012*). Spt5 proteins contain a NusG-like N-terminal (NGN) domain and one (archaea) or several (eukaryotes) KOW domains (*Werner, 2012*). All structurally characterized NusG/Spt5-KOW domains adopt a five-stranded β-barrel structure (*Figure 1A*; see e.g. *Klein et al., 2011*; *Meyer et al., 2015*; *Mooney et al., 2009*; *Zuber et al., 2018*).

*Ec*RfaH is an operon-specific paralog of *E. coli* NusG (*Ec*NusG) and – just like *Ec*NusG – consists of an NGN domain that is loosely connected to a KOW domain via a flexible 15 amino acid long linker. However, in free *Ec*RfaH *Ec*RfaH-KOW folds as an α-helical hairpin (all-α state) that interacts with the *Ec*RfaH-NGN domain. Thus, the binding site for RNA polymerase (RNAP) at the domain interface on *Ec*RfaH-NGN is masked and *Ec*RfaH is locked in an autoinhibited state (*Belogurov et al., 2007*). Upon recruitment to a transcription elongation complex pausing at an *operon polarity suppressor* (*ops*) site, *Ec*RfaH is activated (*Artsimovitch and Landick, 2002*; *Zuber et al., 2019*): the domains dissociate and the liberated *Ec*RfaH-KOW refolds into a NusG-KOW-like β-barrel (all-β state; *Figure 1A and B*; *Burmann et al., 2012*; *Zuber et al., 2019*).

The refolding occurs spontaneously as soon as the domains are separated and *Ec*RfaH-KOW, when produced as an isolated domain, also adopts the all-β state, implying that the all-α fold is only stable in the presence of *Ec*RfaH-NGN (*Burmann et al., 2012*; *Tomar et al., 2013*). Each of the *Ec*RfaH-KOW states has a specific function: the all-α state prevents off-target recruitment of *Ec*RfaH and competition with the general transcription factor NusG (*Belogurov et al., 2007*), whereas the all-β *Ec*RfaH-KOW serves as recruitment platform for ribosomes to activate translation (*Burmann et al., 2012*; *Zuber et al., 2019*). Upon release from RNAP *Ec*RfaH is transformed back into its autoinhibited state, that is, the structural switch of *Ec*RfaH-KOW is fully reversible (*Zuber et al., 2019*). *Ec*RfaH was not only considered a fold-switching protein, but termed a 'transformer protein' to emphasize, that a complete domain cycles reversibly between two states with radically different stable secondary/tertiary structure and with each state performing a distinct function (*Knauer et al., 2012*).

The fine-tuned mechanism used by *Ec*RfaH to control its functions may be widespread in nature (*Porter and Looger, 2018*) and a recent study predicts that 24% of the bacterial NusG family members might exhibit similar reversible α-to-β transitions (*Porter et al., 2022*). However, the molecular principles underlying the fold-switching process are only poorly understood. Here, we present a comprehensive thermodynamic and structural analysis of six KOW domains from NusG/Spt5/RfaH proteins from all domains of life. We combine circular dichroism (CD) spectroscopy, differential scanning calorimetry (DSC), and solution-state nuclear magnetic resonance (NMR) spectroscopy to gain insight into the mechanism and the dynamics of fold-switching within the RfaH family on a molecular level and provide a rationale for the mechanism of fold-switching proteins in general.

## Results
### Evolutionary conservation of fold-switching within the RfaH family

To date, three-dimensional structures and comprehensive evidence for fold-switching are available only for *Ec*RfaH (*Belogurov et al., 2007*; *Burmann et al., 2012*; *Zuber et al., 2019*), although other RfaH orthologs seem to employ a similar mechanism to carry out their function (*Carter et al., 2004*; *Porter et al., 2022*). Thus, we first asked whether this ability might be a general feature of RfaH proteins. We chose RfaH from *Vibrio cholerae* (*Vc*RfaH) for a structural analysis by solution-state NMR spectroscopy as it is evolutionarily remote from *Ec*RfaH (sequence identity *Ec*/*Vc*RfaH: 43.6% [full-length] or 35.8% [KOW domain], respectively). We first identified the secondary structure elements of the full-length protein by performing an NMR backbone assignment and calculating the secondary chemical shift for each $^{13}C\alpha$ and $^{13}CO$ atom, which depends on the main chain geometry (*Figure 1C*). In full-length *Vc*RfaH, the KOW domain exhibits two stretches with helical structure that are separated by about four residues and the overall pattern of secondary structure elements perfectly matches the one of auto-inhibited *Ec*RfaH (*Burmann et al., 2012*), suggesting similar tertiary structures for *Ec*RfaH and *Vc*RfaH (compare *Figure 1A*), but with helix $\alpha_3^*$ being 1.5 turns longer in *Vc*RfaH. Interestingly, the Cα and Cβ atoms of C34 and C102 exhibit chemical shifts typical for cystines (*Sharma and Rajarathnam, 2000*, *Figure 1—figure supplement 1A*). These residues are located at the end of helix $\alpha_3^*$ and in strand $\beta_3^*$, respectively, and are, most probably, in close proximity, as indicated by the structure of *Ec*RfaH.

**Table 1.** Solution structure statistics for *Vc*RfaH-KOW.

| **Experimental derived restraints** | | | |
|---|---|---|---|
| Distance restraints | | | |
| | | NOEs unique (total) | 630 (734) |
| | | Intraresidual | 59 |
| | | Sequential | 187 |
| | | Medium range | 89 |
| | | Long range | 295 |
| | | Hydrogen bonds | 2 · 18 |
| | | | |
| Dihedral restraints | | | 76 |
| Restraint violation | | | |
| Average distance restraint violation (Å) | | | 0.002584±0.000700 |
| Maximum distance restraint violation (Å) | | | 0.12 |
| Average dihedral restraint violation (°) | | | 0.0654±0.0265 |
| Maximum dihedral restraint violation (°) | | | 0.71 |
| Deviation from ideal geometry | | | |
| Bond length (Å) | | | 0.000544±0.000039 |
| Bond angle (Å) | | | 0.1096±0.0056 |
| Coordinate precision*,† | | | |
| Backbone heavy atoms (Å) | | | 0.32 |
| All heavy atoms (Å) | | | 0.90 |
| Ramachandran plot statistics‡ (%) | | | 91.8/7.9/0.2/0.1 |

*The precision of the coordinates is defined as the average atomic root mean square difference between the accepted simulated annealing structures and the corresponding mean structure calculated for the given sequence region.

†Calculated for residues 116–165.

‡Ramachandran plot statistics are determined by PROCHECK and noted by most favored/ additionally allowed/ generously allowed/disallowed.

The addition of a reducing agent to [$^2$H, $^{15}$N, $^{13}$C]-*Vc*RfaH led to drastic changes of the chemical shifts of C34 and C102 as well as residues in spatial proximity in a [$^1$H, $^{15}$N]-heteronuclear single quantum coherence (HSQC) spectrum (*Figure 1—figure supplement 1B*). From this we conclude that C34 and C102 form a disulfide bridge, that covalently tethers the α$_3$*-helix to the core of *Vc*RfaH-NGN, a feature absent in *Ec*RfaH. However, upon refolding from a solution containing 8 M urea and reducing agent, $^{15}$N-*Vc*RfaH adopted the same conformation as before denaturation (*Figure 1—figure supplement 1C*), suggesting that the disulfide bridge is not required for *Vc*RfaH to fold into the autoinhibited state.

Next, we determined the solution structure of the isolated *Vc*RfaH-KOW domain by NMR spectroscopy. *Vc*RfaH-KOW also shows the five-stranded β-barrel topology typical for NusG/Spt5-KOW domains (*Figure 1D* and *Table 1*), with a Cα root mean square deviation (rmsd) of 1.4 Å as compared to isolated *Ec*RfaH-KOW.

Although we do not present functional data on *Vc*RfaH here, these results strongly suggest that *Vc*RfaH-KOW can also switch between an all-α and an all-β state and that *Vc*RfaH thus is, most probably, also a transformer protein.

## The model systems

The sequence of NusG/Spt5-KOW domains has been evolutionarily optimized to fold in only one defined conformation. Consequently, in the case of RfaH-KOW, the ability to switch between the all-α

and the all-β state must be encoded within the primary structure, whereas the 'decision' which state to adopt solely depends on the availability of RfaH-NGN (*Tomar et al., 2013*). Sequence alignments and bioinformatical approaches (*Balasco et al., 2015*; *Bernhardt and Hansmann, 2018*; *Gc et al., 2014*; *Joseph et al., 2019*; *Li et al., 2014*; *Shi et al., 2017*; *Xiong and Liu, 2015*) gave first hints why RfaH, in contrast to NusG, is a metamorphic protein and how the structural switch might proceed. Yet, experimental evidence is still scarce. Thus, we analyzed isolated KOW domains of six NusG/Spt5 or RfaH proteins to identify characteristic properties of fold-switching proteins and to understand the molecular mechanisms underlying the refolding mechanism of RfaH-KOW. Due to the fact that NusG proteins are universally conserved, we chose NusG-KOWs from *E. coli* and *Mycobacterium tuberculosis* (*Ec/Mt*NusG-KOW), the Spt5-KOW from the hyperthermophilic archaeon *Methanocaldococcus jannaschii* (*Mj*Spt5-KOW) and the fifth KOW domain from human Spt5 (hSpt5-KOW5) as representative NusG-/Spt5-KOWs and the *Ec/Vc*RfaH-KOWs as representatives for RfaH proteins. The constructs used are about 65 residues in length and contain the structured region and parts of the neighboring linker(s) (*Figure 1—figure supplement 2A*). In isolation all six domains exhibit the typical β-barrel topology (*Figure 1—figure supplement 2B*) with major differences only in the loops or turns connecting the β-strands (*Figure 1—figure supplement 2C*).

## Thermal and chemical stability of the KOW domains

Metamorphic proteins that switch between two stable conformations are expected to show two main minima in their energy landscape, each corresponding to one of these states (*Dishman and Volkman, 2018*). This implicates that (i) in order to control the structural interconversion, one of the conformations has to be (de)stabilized according to a molecular signal, and (ii) the energy minima cannot be as deep as the global minimum of a protein with a single, stable conformation to avoid permanent trapping of one state. Consequently, the all-β RfaH-KOW should show a limited thermodynamic stability to allow facile refolding to the all-α state when RfaH-NGN is available after transcription termination. To test this hypothesis, we analyzed the thermal stability of the six KOW domains by CD-based thermal denaturation experiments (*Figure 2A*) and by DSC (*Figure 2B*) at pH 4 and pH 7. At pH 7 unfolding was reversible for all KOW domains except for hSpt5-KOW5, which showed aggregation; the opposite effect was observed at pH 4 (*Figure 2—figure supplement 1*). All observed unfolding transitions were analyzed with a two-state model to determine the melting temperature, $T_m$, the enthalpy of unfolding at $T_m$, $\Delta H_u(T_m)$, and, in case of the DSC thermograms, the temperature-dependent difference in heat capacity between the N and U states, $\Delta C_p(T)$ (*Figure 2C and D* and *Table 2*).

Due to the fact that the KOW domains are β-barrels the precision of the thermodynamic parameters determined by CD spectroscopy is not as high as for proteins with helical elements. Nevertheless, the results obtained by DSC and CD spectroscopy are in good agreement showing that *Ec*NusG-KOW, *Mt*NusG-KOW, and *Mj*Spt5-KOW have much higher $T_m$ values (87°C, 77°C, and 111°C, respectively) than hSpt5-KOW5 (58–60°C), *Ec*RfaH-KOW (47–50°C), and *Vc*RfaH-KOW (65–70°C). The same trend was observed for $\Delta H_u(T_m)$ values. Consequently, this data indicates that *Ec*NusG-KOW, *Mt*NusG-KOW, and *Mj*Spt5-KOW have a higher thermodynamic stability than Spt5-KOW5, *Ec*RfaH-KOW, and *Vc*RfaH-KOW.

To corroborate and complement the previous findings, we next performed far-UV CD-based chemical unfolding experiments at pH 4 and pH 7 using urea as denaturant (*Figure 3A–F*, left).

EcNusG-KOW, *Mt*NusG-KOW, hSpt5-KOW5, and *Vc*RfaH-KOW show a sigmoidal unfolding curve at either pH, indicative of a two-state unfolding process. Analysis of this data by the linear extrapolation model yields transition midpoints ([urea]$_{1/2}$ values) and $\Delta G_u(H_2O)$ values that confirm the relative order of the stability as determined by thermal denaturation (*Figure 3G and H*, *Table 3*, and *Figure 2C and D*). For *Mj*Spt5-KOW only the native state baseline is observable at both pH values, demonstrating that no denaturation could be achieved and that, consequently, this KOW domain exhibits the highest thermodynamic stability (assuming an *m* value comparable to that of the other KOW domains, MjSpt5-KOW likely has a $\Delta G_u(H_2O)$ value >30–40 kJ/mol). Notably, we obtained a $\Delta G_u(H_2O)$ value for hSpt5-KOW5 at pH 7, showing that this domain has a stability comparable to that of *Vc*RfaH-KOW at physiological pH (*Table 3*). As *Vc*RfaH-KOW, in contrast to all other KOW domains in this study, contains a Trp residue an additional fluorescence-based denaturation experiment was performed, and the obtained parameters are in good agreement with the CD data (*Table 3* and *Figure 3—figure supplement 1A*).

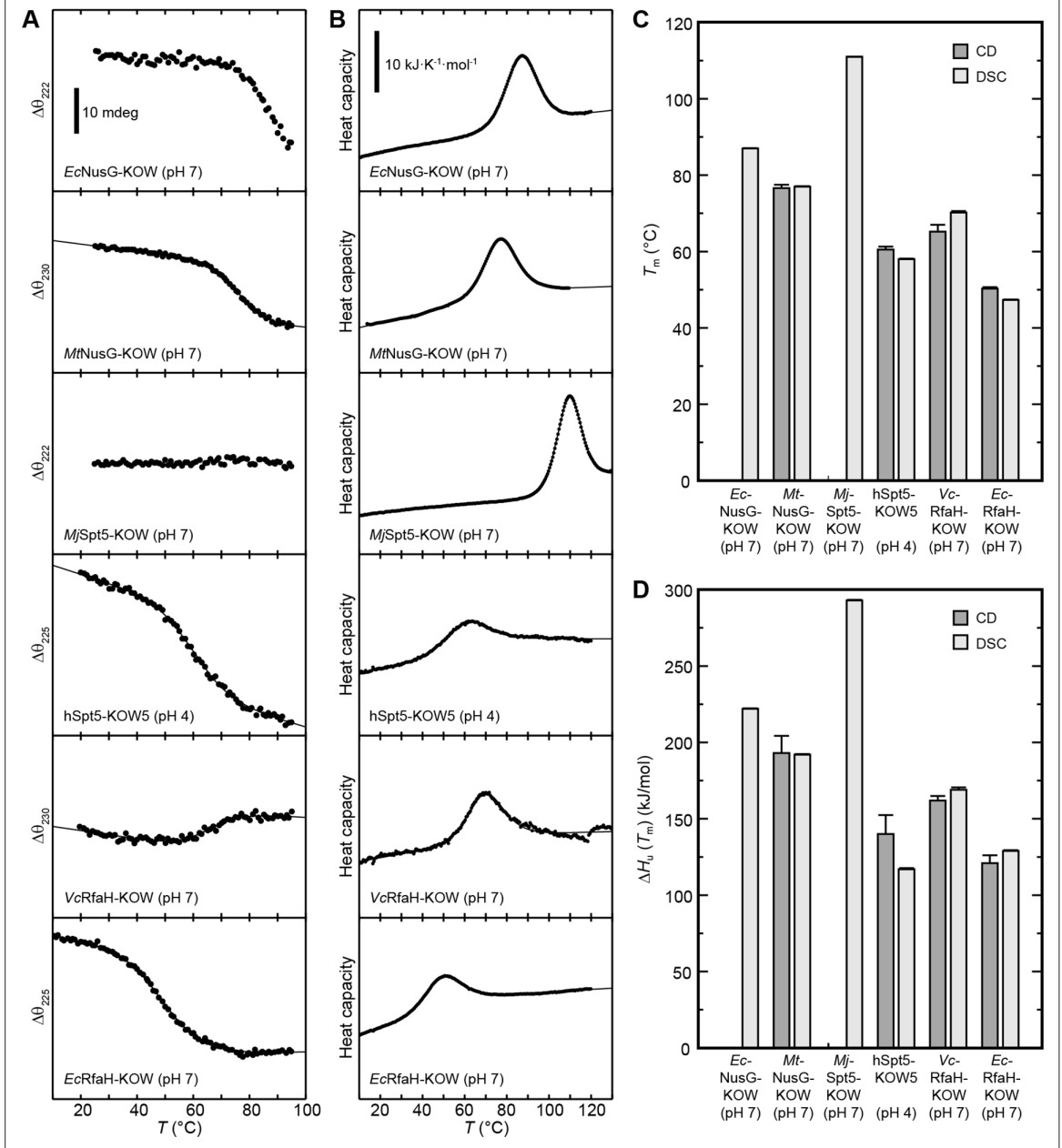

**Figure 2.** Thermal unfolding experiments of the six Kyrpides, Ouzounis, Woese (KOW) domains. (**A**) Thermal unfolding monitored via change in the circular dichroism (CD) signal with a temperature gradient from 20°C to 95°C. The line corresponds to the best fit to a two-state unfolding model. Measurements were carried out with proteins in 10 mM K-acetate (pH 4.0) buffer for hSpt5-KOW5 and in 10 mM K-phosphate (pH 7.0) buffer for all other domains. The wavelength for monitoring the transition was chosen based on the largest difference between the spectra of the folded and unfolded protein (for details, see Materials and methods). Data for *Ec*NusG-KOW was not fitted due to the lack of the baseline of the unfolded state. *Mj*Spt5-KOW could not be denatured at all. (**B**) Thermograms obtained from differential scanning calorimetry (DSC) measurements. All profiles are normalized to one molar of protein. The lines correspond to best fits to a two-state unfolding model that includes a $T$-dependent $\Delta C_p$ change. Buffers are as in (**A**). (**C,D**) $T_m$ (**C**) and $\Delta H_u(T_m)$ (**D**) values derived from thermal unfolding experiments monitored by CD and DSC. Error bars result from data fitting.

The online version of this article includes the following source data and figure supplement(s) for figure 2:

**Source data 1.** Data for thermal denaturation experiments for all Kyrpides, Ouzounis, Woese (KOW) domains.

**Figure supplement 1.** Reversibility of thermal unfolding.

**Table 2.** Selected thermodynamic parameters of the six Kyrpides, Ouzounis, Woese (KOW) domains.

The values were derived from thermal denaturations monitored by differential scanning calorimetry (DSC) and circular dichroism (CD) spectroscopy. Standard deviations result from data fitting.

| Parameter | *Ec*NusG-KOW | *Mt*NusG-KOW | *Mj*Spt5-KOW | hSpt5-KOW5 | *Ec*RfaH-KOW | *Vc*RfaH-KOW |
|---|---|---|---|---|---|---|
| $T_m$ (°C) pH 7/pH 4 | | | | | | |
| CD | –*/– | 76.6±0.874/– | –†/–† | –/60.5±0.771 | 50.3±0.388/– | 65.2±1.78/– |
| DSC | 87.0±0.0485/– | 77.0±0.0885/– | 111±0.0326/– | –/58.0±0.162 | 47.3±0.143/– | 70.2±0.379/– |
| $\Delta H_u$ ($T_m$) (kJ/mol) pH 7/pH 4 | | | | | | |
| CD | –*/– | 193±11.3/– | –†–/–† | –/140±12.4 | 121±5.15/– | 162±2.91/– |
| DSC | 222±0.339/– | 192±0.417/– | 293±0.345/– | –/117±0.735 | 129±0.432/– | 169±1.56/– |
| $\Delta C_p$ ($T_m$) (kJ/(K mol)) pH 7/pH 4 | 0.800/– | 0.346/– | –*/– | –/2.27 | 2.18/– | 0.148/– |

*Data was not fitted due to the lack of the baseline of the unfolded state.
†No denaturation.

To complement the analysis, we repeated the unfolding experiments at pH 7 using guanidinium chloride (GdmCl; *Figure 3A-F*, right, *Table 3* and *Figure 3—figure supplement 1B*). As GdmCl is a more potent denaturant than urea, we were now able to denature even *Mj*Spt5-KOW, giving a [GdmCl]$_{1/2}$ value of 5.03 M, which is more than twice the value of the next stable protein. In accordance with the urea-based unfolding experiments at pH 7, *Mj*Spt5-KOW, *Ec*NusG-KOW, and *Mt*NusG-KOW exhibit higher $\Delta G_u$(H$_2$O) and [denat]$_{1/2}$ values than *Vc*RfaH-KOW and hSpt5-KOW5, although the relative order of stability of *Mt*NusG-KOW and *Ec*NusG-KOW is swapped. This difference as well as the difference between the absolute $\Delta G_u$(H$_2$O) values derived from the urea- and GdmCl-based denaturations is a well-documented phenomenon and may be attributed to the limited applicability of the linear extrapolation model for the analysis of denaturations by GdmCl (see e.g. *Gupta et al., 1996*; *Makhatadze, 1999*). Thus, we base our conclusions on the relative comparison of the obtained values. We finally note that chemical unfolding was completely reversible in all cases (*Figure 3— figure supplement 2*).

Surprisingly, and in contrast to all other domains, *Ec*RfaH-KOW shows a more complex unfolding curve in both urea- and GdmCl-based denaturation experiments at pH 7, with an additional plateau at ≈3 M urea or ≈1 M GdmCl, respectively, between the N and U baselines (*Figure 3F*; no curve could be obtained at pH 4 due to native state aggregation). This suggests that the unfolding of *Ec*RfaH-KOW may be described via a three-step model including an observable equilibrium intermediate that might play an important role in the fold-switching mechanism of *Ec*RfaH-KOW.

In summary, the poor spectroscopic properties of the analyzed domains limit the precision of the absolute values of the thermodynamic parameters obtained from CD experiments. However, our findings reveal clear differences in the global stability of the six domains and allow a grouping into two classes: *Mj*Spt5-KOW and *Ec*/*Mt*NusG-KOW are considered as 'stable domains', whereas the β-barrel *Ec*/*Vc*RfaH-KOW as well as hSpt5-KOW5 show a reduced thermodynamic stability.

## Regions that are unfolded in all-α RfaH-KOW are destabilized in the all-β conformation

We next asked whether the less stable KOW domains also exhibit local differences in their stability as compared to the NusG-KOWs and *Mj*Spt5-KOW. Therefore, we identified the backbone H-bond pattern in the six domains and quantified the magnitude of the through H-bond coupling constant, $^{h3}J_{NC'}$, by long-range HNCO NMR experiments (*Table 4*). This parameter is inversely proportional to the length of the H-bond and the deviation from its optimum angle, thus reflecting the H-bond strength (*Grzesiek et al., 2004*). To allow comparison between the six domains, we grouped H-bonds that are located at equivalent positions of the β-barrels and ordered them according to their position in the individual β-sheets (*Figure 4A and B*).

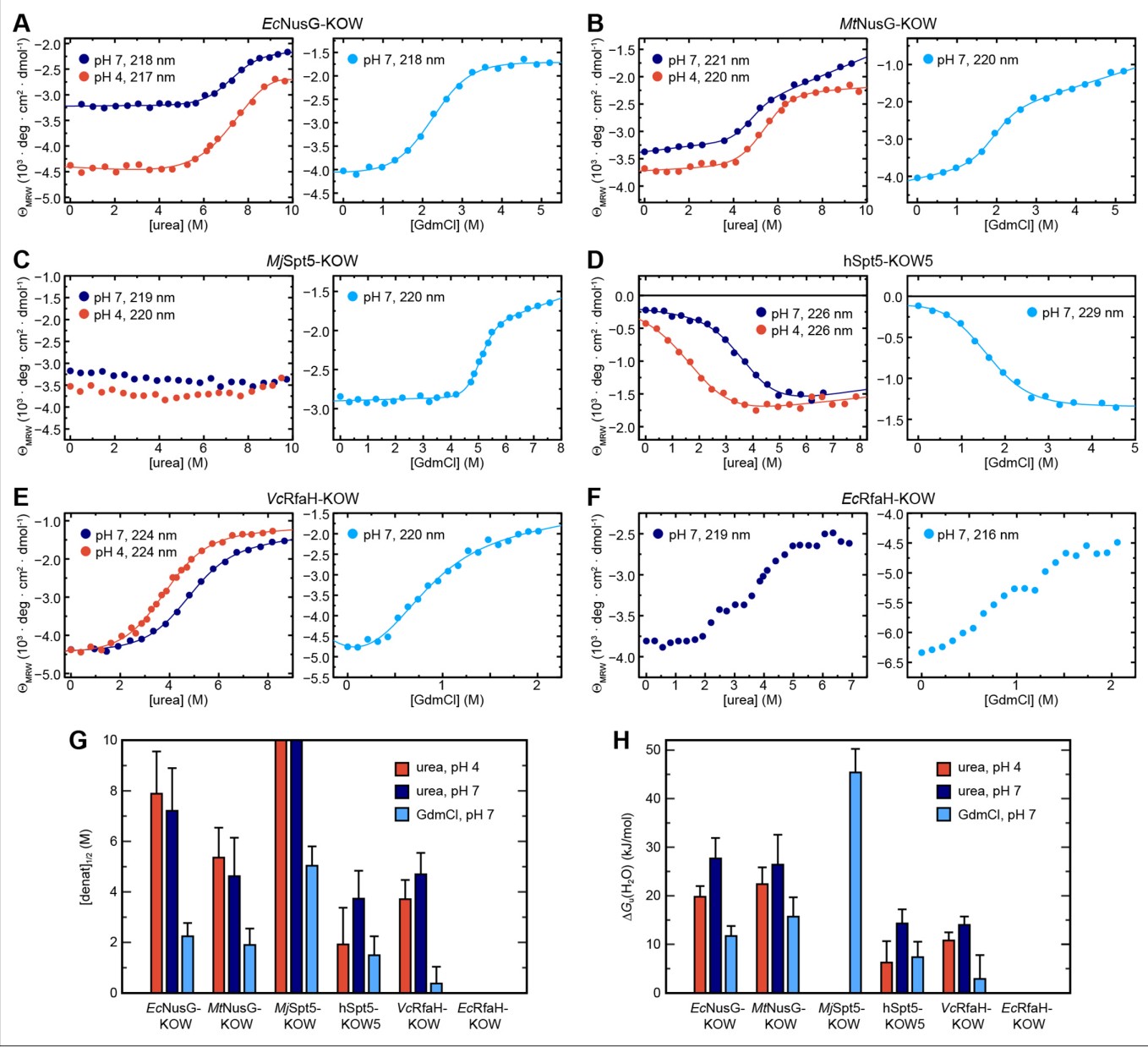

**Figure 3.** Circular dichroism (CD) spectroscopy-based chemical equilibrium unfolding of the six Kyrpides, Ouzounis, Woese (KOW) domains. (**A–F**) Change in $\Theta_{MRW}$ of the indicated protein domain upon over-night incubation with increasing concentrations of (left) urea in 10 mM K-acetate (pH 4.0; red circles) or 10 mM K-phosphate (pH 7.0; dark blue circles), respectively, and (right) GdmCl in 10 mM K-phosphate (pH 7.0; light blue circles). The detection wavelength is indicated and chosen based on the maximum difference between the spectra of the folded and unfolded state (for details see Materials and methods). The lines correspond to the best fits to a two-state unfolding model, except for *Ec*RfaH-KOW, which exhibits a three-state unfolding behavior. (**G, H**) Comparison of [denat]$_{1/2}$ values (**G**) and $\Delta G_u$(H$_2$O) values (**H**) of the KOW domains derived from the chemical denaturation experiments shown in (**A–E**). Error bars result from data fitting.

The online version of this article includes the following source data and figure supplement(s) for figure 3:

**Source data 1.** Data for chemical denaturation experiments for all Kyrpides, Ouzounis, Woese (KOW) domains.

**Figure supplement 1.** Chemical unfolding of *Vc*RfaH-KOW monitored by change in Trp fluorescence.

**Figure supplement 2.** Reversibility of chemical denaturation.

**Table 3.** Thermodynamic parameters of the six Kyrpides, Ouzounis, Woese (KOW) domains.
The values were derived from chemical denaturations monitored by circular dichroism (CD) spectroscopy as well as fluorescence spectroscopy where indicated. Standard deviations result from data fitting.

| Parameter | *Ec*NusG-KOW | *Mt*NusG-KOW | *Mj*Spt5-KOW | hSpt5-KOW5 | *Ec*RfaH-KOW | *Vc*RfaH-KOW |
|---|---|---|---|---|---|---|
| $\Delta G_u(H_2O)$ (25°C) (kJ/mol) | | | | | | |
| Urea, pH 4 | 19.8±2.21 | 22.4±3.46 | –* | 6.24±4.42 | – (native state aggregation) | 10.8±1.66 (10.8±0.90)† |
| Urea, pH 7 | 27.7±4.21 | 26.4±6.16 | –* | 14.3±2.90 | Three-state | 14.0±1.74 (13.9±0.61)† |
| GdmCl, pH 7 | 11.7±2.07 | 15.7±3.99 | 45.4±4.83 | 7.37±3.16 | Three-state | 2.87±4.92 (2.84±6.55)† |
| *m* (25°C) (kJ/(mol M)) ‡ | | | | | | |
| Urea, pH 4 | 2.51±0.453 | 4.18±0.660 | –* | 3.25±0.857 | – (native state aggregation) | 2.91±0.396 (2.98±0.22)† |
| Urea, pH 7 | 3.84±0.681 | 5.71±1.32 | –* | 3.83±0.820 | Three-state | 2.98±0.388 (3.13±0.14)† |
| GdmCl, pH 7 | 5.22±0.809 | 8.26±1.87 | 9.02±0.984 | 4.95±1.31 | Three-state | 7.71±3.68 (7.86±4.29)† |
| [Denat]$_{1/2}$ (25°C) (M) | | | | | | |
| Urea, pH 4 | 7.89 | 5.36 | >10* | 1.92 | – (native state aggregation) | 3.71 (3.62)† |
| Urea, pH 7 | 7.21 | 4.62 | >10* | 3.73 | ~2.25/~4.25 | 4.70 (4.44)† |
| GdmCl, pH 7 | 2.24 | 1.90 | 5.03 | 1.49 | ~0.6/~1.3 | 0.37 (0.36)† |

*No denaturation possible.

†Values were determined by fluorescence-based unfolding experiments.

‡The *m* value is a measure of the broadness of the transition and correlates with the difference in the accessible surface area between N and U, and the transition midpoint.

Most $|^{h3}J_{NC'}|$ values are in the range of 0.5–0.9 Hz, which is typical for H-bonds of β-sheets (***Grzesiek et al., 2004***). In line with having the highest $T_m$, *Mj*Spt5-KOW often exhibits the highest coupling constants, which is indicative of a highly rigid packing of the β-barrel. Strikingly, *Mj*Spt5-KOW has three additional H-bonds between strands β$_5$ and β$_1$ (#22–24), which provides an extra stabilization of the C-terminal β-strand that may contribute to the high thermostability of this protein. The 'stable' domains (i.e. *Ec*/*Mt*NusG-KOW and *Mj*Spt5-KOW) show their strongest H-bonds in two regions, namely between strands β$_1$:β$_2$ and β$_3$:β$_4$. In addition, most of these H-bonds are more stable than corresponding H-bonds in *Ec*/*Vc*RfaH-KOW and hSpt5-KOW5, implying that the H-bonds in the domains with reduced stability are more dynamic and on average longer or involve a less optimal bonding angle. From this we conclude that in *Ec*/*Vc*RfaH-KOW and hSpt5-KOW5 strands β$_1$ and parts of β$_4$ are less stably bound to the rest of the β-barrel than in the stable domains. Moreover, together with the fact that β$_1$, the C-terminal half of β$_4$, and β$_5$ are disordered in the all-α state of the *Ec*/*Vc*RfaH-KOW (***Figure 4C***), this also reflects the chameleonic folding behavior of these regions in the all-β state.

## hSpt5-KOW5, *Ec*- and *Vc*RfaH-KOW exchange with a globally unfolded conformer on the ms time scale

To assess the folding mechanism of the KOW domains at the amino acid level, we performed an NMR-based analysis of the structural dynamics of the six β-barrel proteins. As larger structural rearrangements, such as folding events, mostly occur at the μs-ms time scale for small proteins or are even slower (***Maxwell et al., 2005***), we focused on the analysis of the slow chemical exchange regime. Therefore, we performed amide $^{15}$N-based chemical exchange saturation transfer (CEST) experiments (***Vallurupalli et al., 2012***). This method allows the sensitive detection and characterization of sparsely populated states (=minor species; relative population $p_B$) that exchange with a major species (relative population $p_A = 1 - p_B$) with a rate $k_{ex}$ of 10–200 s$^{-1}$. The detection is achieved by frequency-selective saturation along the $^{15}$N dimension that is 'transferred' from the minor to the major species. This decreases the signal intensity of the major species and then leads to an additional dip in the CEST

**Table 4.** Quantification of H-bond strengths from LR-HNCO nuclear magnetic resonance (NMR) experiments for all Kyrpides, Ouzounis, Woese (KOW) domains.

| H-bond # | β-Sheet | EcNusG-KOW Donor | Acceptor | $|^{h3}J_{NC}|$ (Hz) | $\sigma\,|^{h3}J_{NC}|$ (Hz) | MtNusG-KOW Donor | Acceptor | $|^{h3}J_{NC}|$ (Hz) | $\sigma\,|^{h3}J_{NC}|$ (Hz) | MjSpt5-KOW Donor | Acceptor | $|^{h3}J_{NC}|$ (Hz) | $\sigma\,|^{h3}J_{NC}|$ (Hz) |
|---|---|---|---|---|---|---|---|---|---|---|---|---|---|
| 1 | β1-β2 | 131 | 148 | 0.69 | 0.0077 | 188 | 205 | 0.61 | 0.0098 | 92 | 109 | 0.70 | 0.013 |
| 2 | β1-β2 | 148 | 132 | 0.72 | 0.0083 | 205 | 189 | 0.69 | 0.0094 | 109 | 93 | 0.79 | 0.0088 |
| 3 | β1-β2 | 134 | 146 | 0.67 | 0.0081 | 191 | 203 | 0.56 | 0.011 | 95 | 107 | 0.77 | 0.012 |
| 4 | β1-β2 | 146 | 134 | 0.62 | 0.0096 | 203 | 191 | 0.62 | 0.0094 | 107 | 95 | 0.67 | 0.0095 |
| 5 | β1-β2 | 136 | 144 | 0.65 | 0.0073 | 193 | 201 | 0.65 | 0.0080 | 97 | 105 | 0.68 | 0.048 |
| 6 | β1-β2 | 143 | 136 | 0.65 | 0.0088 | 200 | 193 | 0.76 | 0.013 | 104 | 97 | 0.82 | 0.012 |
| 7 | β2-β3 | 147 | 161 | 0.37 | 0.0105 | 204 | 218 | 0.31 | 0.015 | 108 | 122 | 0.54 | 0.010 |
| 8 | β2-β3 | 161 | 147 | Peak overlap | | 218 | 204 | 0.69 | 0.011 | 122 | 108 | 0.65 | 0.010 |
| 9 | β2-β3 | 149 | 159 | 0.67 | 0.012 | 206 | 216 | 0.53 | 0.013 | 110 | 120 | 0.57 | 0.030 |
| 10 | β2-β3 | 159 | 150 | 0.50 | 0.0086 | 216 | 207 | 0.45 | 0.012 | 120 | 111 | 0.60 | 0.010 |
| 11 | β2-β3 | 152 | 157 | 0.46 | 0.019 | 209 | 214 | Peak overlap | | 113 | 118 | – | – |
| 12 | β3-β4 | 158 | 173 | 0.78 | 0.0077 | 215 | 230 | 0.83 | 0.0088 | 119 | 134 | No HNCO peak | |
| 13 | β3-β4 | 173 | 158 | 0.64 | 0.010 | 230 | 215 | 0.62 | 0.011 | 134 | 119 | 0.62 | 0.010 |
| 14 | β3-β4 | 160 | 171 | 0.73 | 0.0056 | 217 | 228 | 0.75 | 0.0089 | 121 | 132 | 0.88 | 0.012 |
| 15 | β3-β4 | 171 | 161 | Peak overlap | | 228 | 217 | 0.73 | 0.0062 | 132 | 121 | 0.47 | 0.014 |
| 16 | β3-β4 | 162 | 169 | 0.51 | 0.0074 | 219 | 226 | 0.50 | 0.0090 | 123 | 130 | No H-bond distance | |
| 17 | β3-β4 | 169 | 162 | 0.60 | 0.0066 | 226 | 219 | 0.61 | 0.0091 | No equivalent | | – | – |
| 18 | β3-β4 | 167 | 164 | – | – | 224 | 221 | 0.20 | 0.019 | No equivalent | | – | – |
| 19 | β5-β1 | 137 | 177 | 0.46 | 0.017 | 194 | 234 | 0.52 | 0.025 | 98 | 138 | No HNCO peak | |
| 20 | β5-β1 | 179 | 135 | Peak overlap | | 236 | 192 | 0.47 | 0.016 | 140 | 96 | Peak overlap | |
| 21 | β5-β1 | 135 | 179 | 0.73 | 0.0060 | 192 | 236 | Peak overlap | | 96 | 140 | Peak overlap | |
| 22 | β5-β1 | 181 | 133 | – | – | 238 | 190 | – | – | 142 | 94 | 0.46 | 0.023 |
| 23 | β5-β1 | No equivalent | | – | – | No equivalent | | – | – | 143 | 94 | 0.27 | 0.022 |
| 24 | β5-β1 | No equivalent | | – | – | No equivalent | | – | – | 94 | 143 | 0.57 | 0.010 |

| H-bond # | β-Sheet | hSpt5-KOW5 Donor | Acceptor | $|^{h3}J_{NC}|$ (Hz) | $\sigma\,|^{h3}J_{NC}|$ (Hz) | EcRfaH-KOW Donor | Acceptor | $|^{h3}J_{NC}|$ (Hz) | $\sigma\,|^{h3}J_{NC}|$ (Hz) | VcRfaH-KOW Donor | Acceptor | $|^{h3}J_{NC}|$ (Hz) | $\sigma\,|^{h3}J_{NC}|$ (Hz) |
|---|---|---|---|---|---|---|---|---|---|---|---|---|---|
| 1 | β1-β2 | 707 | 724 | 0.60 | 0.0074 | 113 | 130 | 0.76 | 0.020 | 116 | 133 | 0.87 | 0.015 |
| 2 | β1-β2 | 724 | 708 | Peak overlap | | 130 | 114 | 0.53 | 0.051 | 133 | 117 | 0.59 | 0.024 |
| 3 | β1-β2 | 710 | 722 | 0.70 | 0.0077 | 116 | 128 | 0.65 | 0.027 | 119 | 131 | Peak overlap | |
| 4 | β1-β2 | 722 | 710 | 0.50 | 0.019 | 128 | 116 | Peak overlap | | 131 | 119 | 0.46 | 0.020 |
| 5 | β1-β2 | 712 | 720 | No HNCO peak | | 118 | 126 | 0.53 | 0.056 | 121 | 129 | 0.57 | 0.014 |
| 6 | β1-β2 | 719 | 713 | Peak overlap | | 125 | 118 | 0.66 | 0.026 | 128 | 121 | 0.62 | 0.017 |

*Table 4 continued on next page*

*Table 4 continued*

| H-bond # | β-Sheet | hSpt5-KOW5 | | | | EcRfaH-KOW | | | | VcRfaH-KOW | | | |
|---|---|---|---|---|---|---|---|---|---|---|---|---|---|
| | | Donor | Acceptor | $|^{h3}J_{NC'}|$ (Hz) | $\sigma\ |^{h3}J_{NC'}|$ (Hz) | Donor | Acceptor | $|^{h3}J_{NC'}|$ (Hz) | $\sigma\ |^{h3}J_{NC'}|$ (Hz) | Donor | Acceptor | $|^{h3}J_{NC'}|$ (Hz) | $\sigma\ |^{h3}J_{NC'}|$ (Hz) |
| 7 | β2-β3 | 723 | 735 | H-bond peak present, but too weak to quantify | | 129 | 142 | 0.41 | 0.029 | 132 | 145 | 0.42 | 0.019 |
| 8 | β2-β3 | 735 | 723 | Peak overlap | | 142 | 129 | 0.70 | 0.019 | 145 | 132 | 0.69 | 0.027 |
| 9 | β2-β3 | 725 | 734 | 0.71 | 0.010 | 131 | 140 | 0.48 | 0.032 | 134 | 143 | 0.61 | 0.036 |
| 10 | β2-β3 | 733 | 726 | – | – | 140 | 132 | 0.96 | 0.021 | 143 | 135 | 1.0 | 0.011 |
| 11 | β2-β3 | 728 | 731 | 0.61 | 0.012 | 134 | 138 | – | – | 137 | 141 | – | – |
| 12 | β3-β4 | 732 | 745 | 0.59 | 0.009 | 139 | 154 | 0.60 | 0.034 | 142 | 157 | Peak overlap | |
| 13 | β3-β4 | 745 | 732 | 0.62 | 0.019 | 154 | 139 | No HNCO peak | | 157 | 142 | 0.68 | 0.015 |
| 14 | β3-β4 | 734 | 743 | 0.69 | 0.039 | 141 | 152 | 0.65 | 0.049 | 144 | 155 | 0.58 | 0.019 |
| 15 | β3-β4 | 743 | 734 | 0.49 | 0.029 | 152 | 141 | – | – | 155 | 144 | 0.72 | 0.021 |
| 16 | β3-β4 | 736 | 741 | 0.63 | 0.033 | 143 | 150 | 0.75 | 0.033 | 146 | 153 | 0.49 | 0.024 |
| 17 | β3-β4 | 741 | 736 | No H-bond orientation | | 150 | 143 | 0.47 | 0.052 | 153 | 146 | 0.48 | 0.015 |
| 18 | β3-β4 | No equivalent | | – | – | 148 | 145 | No H-bond orientation | | 151 | 148 | No HNCO peak | |
| 19 | β5-β1 | 713 | 749 | No HNCO peak | | 119 | 158 | Peak overlap | | 122 | 161 | – | – |
| 20 | β5-β1 | 751 | 711 | – | | 160 | 117 | 0.51 | 0.036 | 163 | 120 | 0.57 | 0.037 |
| 21 | β5-β1 | 711 | 751 | 0.68 | 0.023 | 117 | 160 | 0.68 | 0.015 | 120 | 163 | 0.53 | 0.016 |
| 22 | β5-β1 | 753 | 709 | – | – | 162 | 115 | – | – | 165 | 118 | – | – |
| 23 | β5-β1 | No equivalent | | – | – | No equivalent | | – | – | No equivalent | | – | – |
| 24 | β5-β1 | No equivalent | | – | – | No equivalent | | – | – | No equivalent | | – | – |

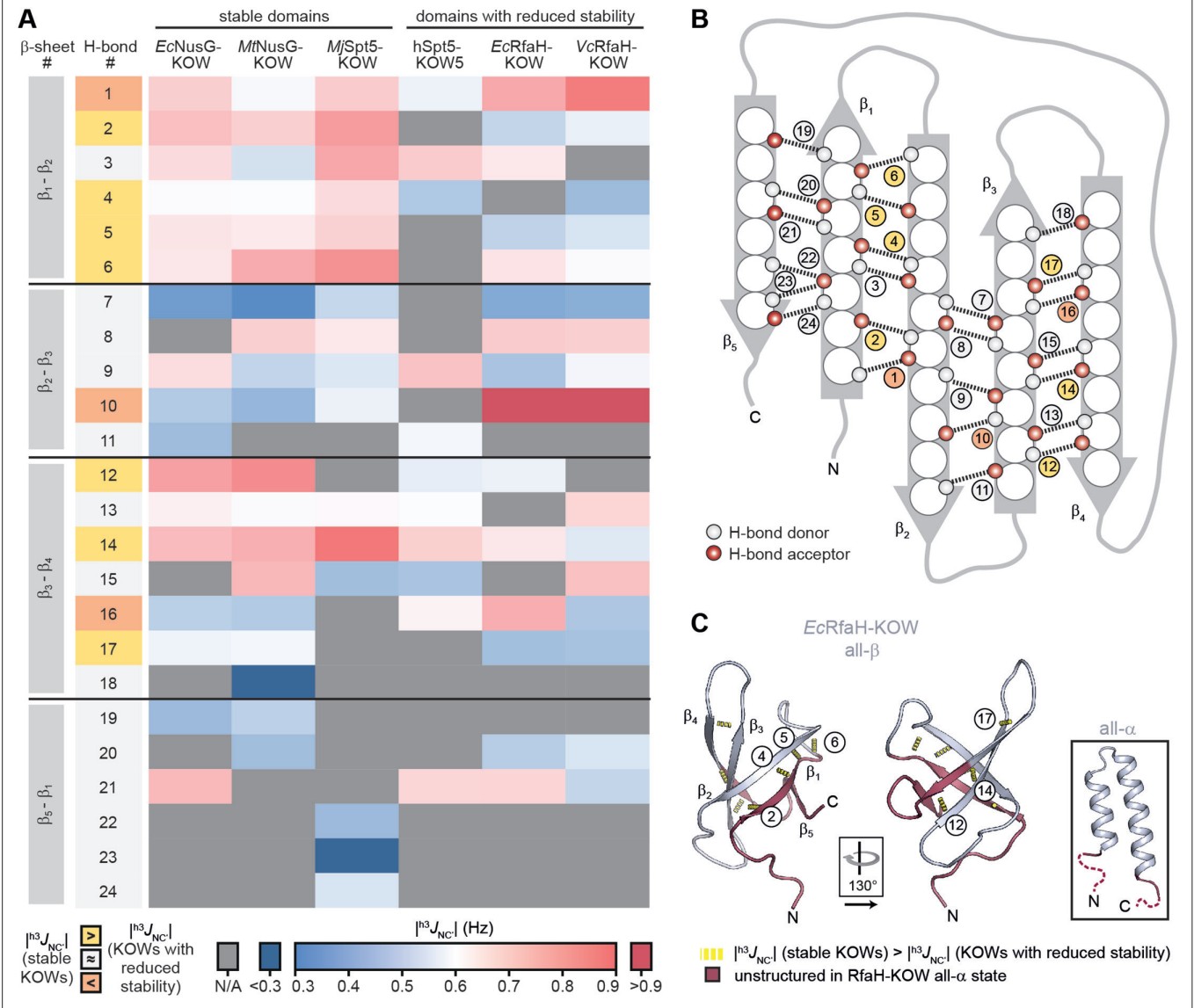

**Figure 4.** H-bond pattern and stability in the six Kyrpides, Ouzounis, Woese (KOW) domains. (**A**) Heat map of the magnitude of the $^{h3}J_{NC'}$ coupling constants of the H-bonds determined by long-range HNCO nuclear magnetic resonance (NMR) experiments. H-bonds that are located at equivalent positions are grouped and ordered according to their location in the respective β-sheet (position within the β-barrel as indicated in (**B**)), and colored according to their $|^{h3}J_{NC'}|$ value as indicated at the bottom. H-bond numbers highlighted in yellow: H-bonds that have lower $|^{h3}J_{NC'}|$ values for at least two of the domains with reduced thermodynamic stability compared to the stable domains; H-bond numbers highlighted in orange: H-bonds that have higher $|^{h3}J_{NC'}|$ values for at least two of the domains with reduced thermodynamic stability compared to the stable domains. (**B**) Scheme of the positions of the H-bonds (dashed lines) within the β-barrel. Amino acids are depicted as spheres. White and red circles represent H-bond donors and acceptors, respectively. H-bonds are color-coded as in (**A**). (**C**) Cartoon representation of all-β EcRfaH-KOW (PDB-ID: 2LCL, gray). Regions that are unstructured in the all-α conformation are colored in dark red. H-bonds that have lower $|^{h3}J_{NC'}|$ values for at least two of the domains with reduced thermodynamic stability compared to the stable domains are shown as yellow dashed tubes and labeled. The relative orientation of the structures is indicated. The inset shows the all-α EcRfaH-KOW (PDB-ID: 5OND; gray; unstructured regions at the termini are colored in dark red and correspond to the dark red regions in the all-β EcRfaH-KOW).

profile (major species signal intensity versus saturation frequency) next to the large major species minimum if there is a difference in the resonance frequencies of the two species.

None of the CEST profiles of EcNusG-KOW, MtNusG-KOW, and MjSpt5-KOW exhibits an exchange peak (**Figure 5A–C**), demonstrating that these domains are stable on the ms time scale, in agreement with their high thermodynamic stabilities (see above).

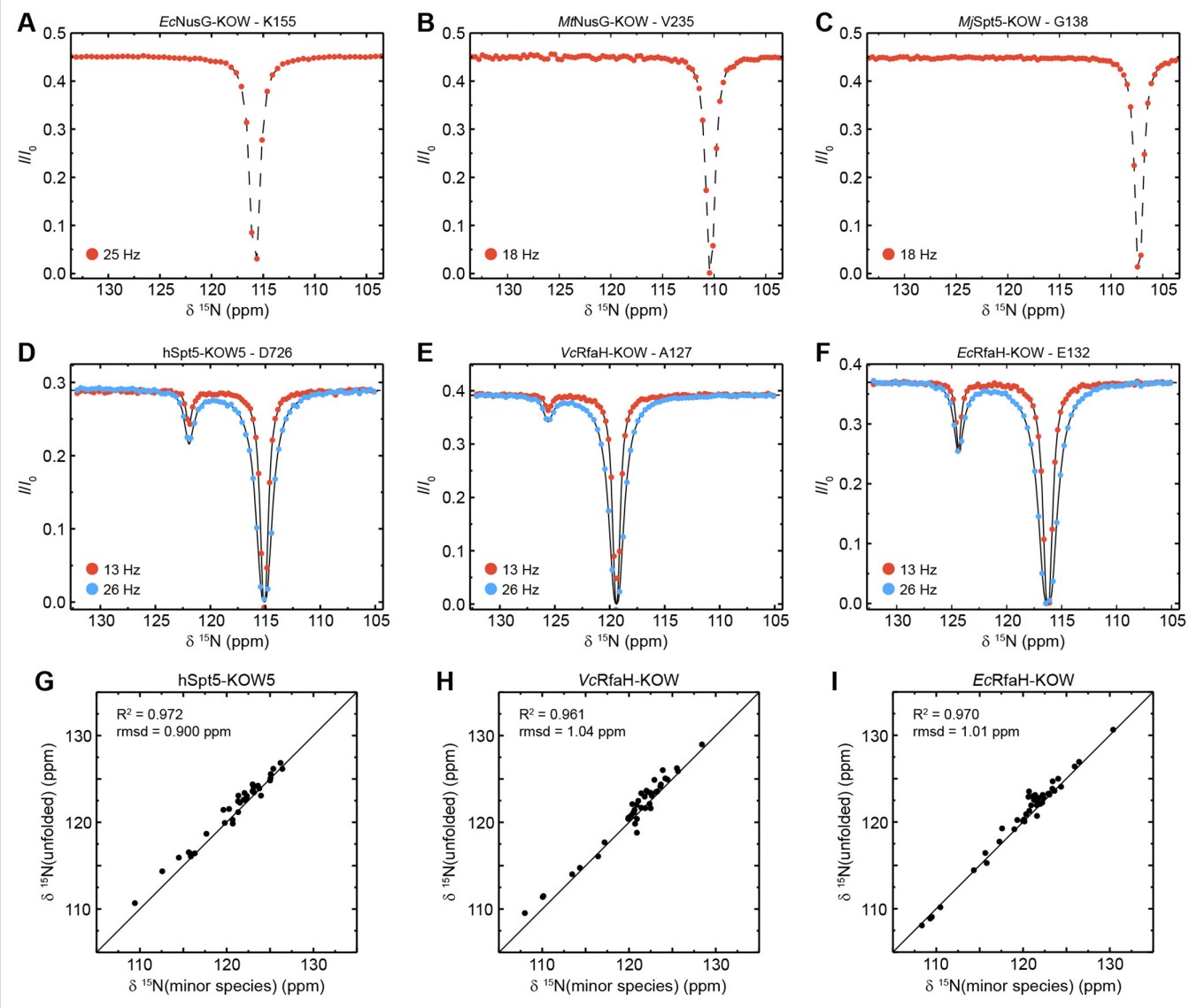

**Figure 5.** Chemical exchange saturation transfer (CEST) analysis of the Kyrpides, Ouzounis, Woese (KOW) domains. (**A–F**) Representative backbone $^{15}$N-CEST profiles of the indicated KOW domain measured with one (**A–C**) or two (**D–F**) $B_1$ field strengths and an exchange time of 0.5 s. $B_0$ field for (**A–C**): 21.15T; $B_0$ field for (**A–C**): 16.45T. The lines in (**D–F**) are fits to a two-state exchange model. (**G–I**) Correlation plots showing the high similarity of the chemical shift of the minor CEST species and that of the corresponding random coil value. The latter were obtained by backbone assignment in 8 M urea (*Ec*RfaH-KOW) or are theoretical values (*Vc*RfaH-KOW, hSpt5-KOW5). The squared correlation coefficient and the root mean square deviation (rmsd) between the two corresponding sets of chemical shifts are listed.

The online version of this article includes the following source data and figure supplement(s) for figure 5:

**Source data 1.** Chemical exchange saturation transfer (CEST) fits for *Ec*RfaH-KOW, *Vc*RfaH-KOW, and hSpt5-KOW5.

**Source data 2.** Experimentally determined chemical shift values of urea-denatured *Ec*RfaH-KOW and predicted random coil chemical shift values of *Vc*RfaH-KOW and hSpt5-KOW5.

**Figure supplement 1.** Extended chemical exchange saturation transfer (CEST) analysis of hSpt5-KOW5, *Vc*RfaH-KOW, or *Ec*RfaH-KOW.

In contrast, most CEST traces of hSpt5-KOW5, *Ec*RfaH-KOW, and *Vc*RfaH-KOW have a second dip, indicating exchange with a second, low-populated state (exemplary traces are shown in *Figure 5D-F*). Using a two-state exchange model, we fitted all CEST traces that showed an exchange signal individually to determine the residue-specific $k_{ex}$ and $p_B$ values. In all three cases, the $k_{ex}/p_B$ values appear to cluster in one region, suggesting a global, cooperative process (*Figure 5—figure supplement 1A*). Thus, we next performed a global fit of all CEST traces for each of the three proteins resulting in global

**Table 5.** Exchange parameters derived from global fitting of the chemical exchange saturation transfer (CEST) experiments to a two-state exchange model.

| Parameter | hSpt5-KOW5 | VcRfaH-KOW | EcRfaH-KOW |
|---|---|---|---|
| $p_A$ (%) | 99.15±0.02 | 99.57±0.01 | 94.47±0.46 |
| $p_B$ (%) | 0.85±0.02 | 0.43±0.01 | 5.53±0.46 |
| $k_{AB}$ (s$^{-1}$) | 0.76±0.03 | 0.32±0.02 | 0.82±0.10 |
| $k_{BA}$ (s$^{-1}$) | 88.62±3.12 | 74.24±3.17 | 13.98±1.24 |
| $k_{ex}$ (s$^{-1}$) | 89.38±3.15 | 74.57±3.18 | 14.80±1.31 |
| $\tau_A$ (s) | 1.31±0.05 | 3.08±0.15 | 1.22±0.15 |
| $\tau_B$ (ms) | 11.28±0.40 | 13.47±0.57 | 71.52±6.33 |
| $\Delta G$ (kJ/mol) | 11.81±0.05 | 13.48±0.07 | 7.18±0.21 |

rate constants and populations as well as lifetimes of the two states (**Table 5**). This analysis yields a relatively high $p_B$ value (5.50%) but low $k_{ex}$ (15.0 s$^{-1}$) for EcRfaH-KOW, a much lower $p_B$ value (0.43%) but higher $k_{ex}$ (75.0 s$^{-1}$) for VcRfaH-KOW, and $p_B/k_{ex}$ values of 0.85% and 89.0 s$^{-1}$ for hSpt5-KOW5.

To characterize the exchanging species structurally, we analyzed the chemical shifts of the minor species. In all three cases, the minor species shifts show a very good correlation with those of a completely unfolded conformation (**Figure 5G–I**; $R^2$ >96%, rmsd <1.04 ppm). Note that the chemical shifts for the unfolded state of EcRfaH-KOW were obtained experimentally by backbone assignment of the protein in 8 M urea, whereas those of VcRfaH-KOW and hSpt5-KOW5 are predicted values (see Materials and methods for details). Determination of the relative populations finally results in the equilibrium constant and the difference in Gibbs free energy, $\Delta G$, separating the energy levels of the two species (**Table 5**). As expected, these $\Delta G$ values are similar to those obtained from the urea-based unfolding experiments at pH 7 (**Table 3**).

Taken together, the CEST experiments show that the folded all-β state of the isolated RfaH-KOWs and also hSpt5-KOW5 is in equilibrium with a species that resembles an unfolded conformation. As this state is easily accessible from the β-barrel, we conclude that the folding barrier separating the two states cannot be too high as this would prohibit an exchange on the ms time scale.

## The unfolded conformers of Ec- and VcRfaH-KOW contain transient helical structures

Although the chemical shifts of the minor species of EcRfaH-KOW nicely correlate with the chemical shifts of the urea-unfolded protein (**Figure 5I**), there are some noticeable differences in the $^{15}$N chemical shifts ($\Delta\delta$ $^{15}$N) of the two data sets (red bars in **Figure 6A**, top panel). In particular, two regions (region 1: Q127–T131, region 2: E136–I150) show significant deviations of −1 to −3 ppm, indicating local residual structures in these regions. As the type of present (secondary) structure cannot be derived from $^{15}$N data, we recorded a CEST experiment on the $^{13}$Cα carbons of $^{13}$C,$^{15}$N-EcRfaH-KOW (**Figure 6—figure supplement 1**) and calculated $\Delta\delta$ $^{13}$Cα between the observed minor species values and the random coil values obtained from the urea-unfolded protein (red bars in **Figure 6A**, bottom panel). The deviations are positive in regions 1 and 2, indicating the presence of helical structures at these sites. This is in agreement with secondary structure predictions, which show that the Leu-rich motif (LLLNL) in region 2, where the deviations of δ $^{15}$N and δ $^{13}$Cα are most pronounced, has high α-helical propensity (**Figure 6—figure supplement 2**; see also **Balasco et al., 2015**). Moreover, the two helical elements are located at the positions of the two α-helices in the all-α form of EcRfaH-KOW (compare **Figure 1B**). Due to the presence of two dips the CEST profiles can be analyzed using a two-state model (minor versus major species). Interestingly, the resulting $^{15}$N transverse relaxation rates ($R_2$ values) of regions 1 and 2 in the minor species are significantly higher than corresponding rates in the β-barrel state (**Figure 6A**, mid panel). Generally, one would expect that the minor species exhibits lower $R_2$ values as it is more flexible due to its largely unfolded nature (**Farrow et al., 1995**). The increased relaxation rates thus indicate the presence of additional exchange processes on the intermediate to fast chemical exchange (i.e. μs–ms) time scale. Consequently, the minor species itself

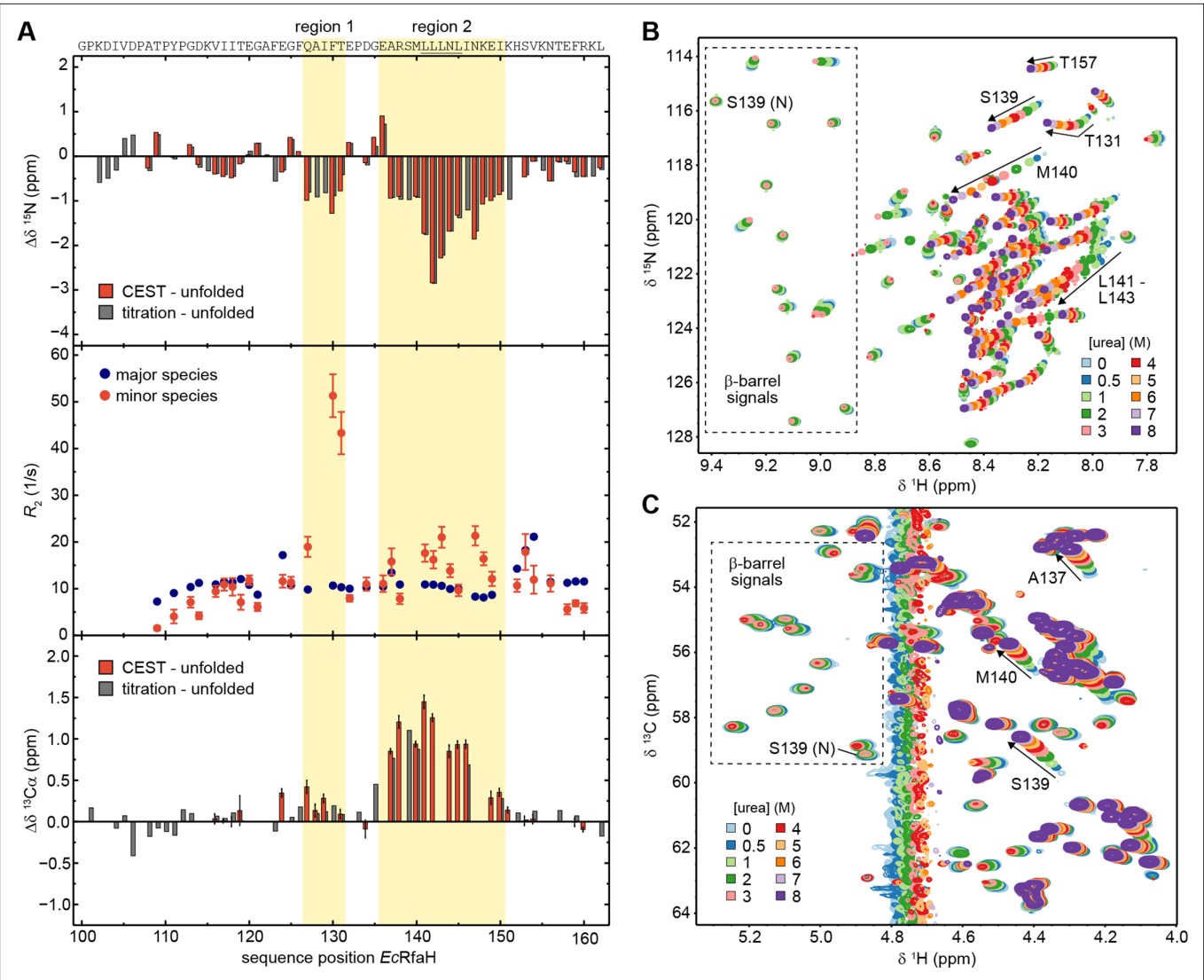

**Figure 6.** The minor species of *Ec*RfaH-KOW contains residual structure. (**A**) Deviations of the minor species of *Ec*RfaH-KOW from the urea-unfolded state. Top row: Sequence-dependent difference between the $^{15}N$ backbone amide chemical shifts of the minor species and of the values obtained by assignment in 8 M urea. The values for the minor species were either obtained from the chemical exchange saturation transfer (CEST) experiment (red bars, individual fits; 'CEST – unfolded') or by tracing back the chemical shift changes from 8 to 0 M urea in the [$^{1}H$, $^{15}N$]-heteronuclear single quantum coherence (HSQC)-based urea titration (gray bars; 'titration – unfolded'; see panel (**B**)). Middle row: $R_2$ values of the major species (*Ec*RfaH-KOW β-barrel; blue) and minor species (red), obtained from fitting the CEST profiles (global fit). Regions 1 and 2 of the minor species have $R_2$ values significantly higher than those of their corresponding β-barrel conformation indicating additional exchange processes, whereas N- and C-terminal regions have $R_2$ values lower than those of their corresponding β-barrel conformation, which is typical for random coil structures. Bottom row: Sequence-dependent difference between the $^{13}C\alpha$ chemical shifts of the minor species and of the values obtained by assignment in 8 M urea. The values for the minor species were either obtained from the CEST experiment (red bars, individual fits; 'CEST – unfolded') or by tracing back the chemical shift changes from 8 to 0 M urea in the [$^{1}H$, $^{13}C$]-ctHSQC-based urea titration (gray bars; 'titration – unfolded'; see panel (**C**)). The sequence of *Ec*RfaH-KOW is given above the diagram, the Leu-rich motif is underlined. Regions 1 and 2 are highlighted. Error bars result from data fitting. (**B, C**) Nuclear magnetic resonance (NMR)-based chemical equilibrium unfolding experiments of *Ec*RfaH-KOW using urea as denaturant. The plots show an overlay of (**B**) [$^{1}H$, $^{15}N$]-HSQC, and (**C**) [$^{1}H$, $^{13}C$]-ctHSQC spectra of [$^{15}N$, $^{13}C$]-*Ec*RfaH-KOW, acquired in the presence of varying urea concentrations. The system was buffered by 20 mM Na-phosphate (pH 6.5), 100 mM NaCl, 1 mM ethylenediaminetetraacetic acid (EDTA), 10% (v/v) D$_2$O. Boxed regions mark signals corresponding to the β-barrel state with the signal of S139 being labeled with 'N' ('native'). Arrows and further labels indicate signals of residues that exhibit strong chemical shift changes in the indirect dimension ($^{15}N$ in (**B**), $^{13}C$ in (**C**)). The spectra are colored as indicated.

The online version of this article includes the following source data and figure supplement(s) for figure 6:

**Source data 1.** 8-Anilino-1-naphthalenesulfonic acid (ANS) binding by *Ec*RfaH-KOW during urea-based denaturation.

**Figure supplement 1.** Exemplary traces of chemical exchange saturation transfer (CEST) experiments recorded on $^{13}C\alpha$ carbons of $^{13}C$-*Ec*RfaH-KOW.

*Figure 6 continued on next page*

*Figure 6 continued*

**Figure supplement 2.** Secondary structure predictions for the six Kyrpides, Ouzounis, Woese (KOW) domains used in this study.

**Figure supplement 3.** The minor species of *Vc*RfaH-KOW contains residual structure.

**Figure supplement 4.** The minor species of hSpt5-KOW5 is completely unfolded.

**Figure supplement 5.** The intermediate state of *Ec*RfaH-KOW is no equilibrium MG.

**Figure supplement 6.** Extended analysis of the urea-induced denaturation of *Ec*RfaH-KOW.

seems to be an ensemble of predominantly unfolded, fast interconverting structures with transient helical elements in regions 1 and 2 rather than a static population. As no dips in addition to the ones of the minor and major species can be observed in the CEST profiles, the population of other states is low and beyond the detection limit of CEST experiments.

Like *Ec*RfaH-KOW, the minor species of *Vc*RfaH-KOW also seems to contain residual structure (***Figure 6—figure supplement 3A***). As the unfolded state of this domain was not assigned experimentally, predicted chemical shift values for the random coil structure were used for the correlation plot (***Figure 5H***). When plotting the $\Delta\delta$ $^{15}$N values versus the sequence position (***Figure 6—figure supplement 3A***), the resulting pattern resembles the one obtained for *Ec*RfaH-KOW (compare ***Figure 6A***, top panel). The regions around residues 103–125 (linker) and 155–165 (C-terminus) show relatively low $\Delta\delta$ $^{15}$N values, indicating a random coil structure, whereas the region around residues 140–150 (corresponding to region 2 in *Ec*RfaH-KOW) exhibits significantly increased $\Delta\delta$ $^{15}$N values, suggesting residual structure, similar to *Ec*RfaH-KOW. However, only very small minor species dips were observed in some traces of a CEST experiment recorded on the $^{13}$Cα carbons of $^{13}$C,$^{15}$N-*Vc*RfaH-KOW (***Figure 6—figure supplement 3B***), which we attribute to the very low population of the *Vc*RfaH-KOW minor species (0.43%) that is at the detection limit of the Cα-CEST experiment (which is less sensitive than the $^{15}$N-CEST). Consequently, we analyzed the CEST profiles only qualitatively. Unambiguous minor species dips could be identified for amino acids predominantly located in the region with residual structure with chemical shifts that are downfield-shifted as compared to random coil values (***Figure 6—figure supplement 3B***), indicating the presence of helical elements. As for *Ec*RfaH-KOW, this is in full agreement with secondary structure predictions, which suggest that all NusG/Spt5-KOW domains adopt four to five β-strands whereas both RfaH-KOW domains exhibit propensities for both β-strands and α-helices, especially in the regions with residual structure in the CEST minor species (***Figure 6—figure supplement 2***). Taken together this data suggests that the *Vc*RfaH-KOW minor species also contains transient residual helical structures.

The hSpt5-KOW5 domain is part of an 'RNA clamp' during transcription elongation in eukaryotes (***Bernecky et al., 2017***) and exhibits the typical β-barrel fold in all available structures. Strikingly, hSpt5-KOW5 also exchanges with an unfolded species under non-denaturing conditions (***Figure 5G***), just as *Ec*RfaH-KOW and *Vc*RfaH-KOW. The magnitude of the differences between the minor species $^{15}$N chemical shifts and the predicted random coil values (***Figure 6—figure supplement 4A***) is similar to that observed for *Vc*RfaH-KOW (***Figure 6—figure supplement 3A***). Interestingly, the minor species' chemical shifts of a $^{13}$Cα -CEST of $^{13}$C,$^{15}$N-hSpt5-KOW5 clearly indicate the absence of any substantial residual structure (***Figure 6—figure supplement 4***). In contrast to all other KOW domains in this study, hSpt5-KOW5 is not located at the very C-terminus of full-length hSpt5, but it is just one out of seven KOW domains being flanked by several hundreds of residues at either terminus. Thus, the stability of this domain may be different in its physiological environment. Taken together, this data suggests that hSpt5-KOW5 is a typical monomorphic β-barrel and that its decreased stability, accompanied by the existence of a minor, unfolded species, may be attributed to the absence of the neighboring domains, although we cannot completely rule out that these features are real, intrinsic properties of hSpt5-KOW5 in the full-length protein with (yet unknown) functional relevance.

As the completely unfolded state was only experimentally assigned for *Ec*RfaH-KOW we will focus on this domain in the following analysis. Owing to its population of 5.5% (***Table 5***), *Ec*RfaH-KOW's minor species should be detectable in standard HSQC spectra, given a sufficiently high signal-to-noise ratio. As we observed a stable intermediate during the CD-based chemical unfolding of *Ec*RfaH-KOW we aimed at analyzing the role of the minor species during the chemical denaturation of *Ec*RfaH-KOW by recording [$^{1}$H, $^{15}$N]- and [$^{1}$H, $^{13}$C]-correlation spectra of [$^{15}$N, $^{13}$C]-labeled *Ec*RfaH-KOW in the presence of various urea concentrations (0–8 M) (***Figure 6B and C***).

In both spectra series, we observed a decrease in peak intensity/volume of the β-barrel signals with increasing urea concentration (boxed regions in *Figure 6B and C*), which is completed at ≈ 4 M urea, indicating that the first transition in the far-UV CD-based chemical denaturation of *Ec*RfaH-KOW (*Figure 3F*) corresponds to the unfolding of its β-barrel (tertiary) structure. This is also corroborated by near-UV CD spectroscopy-based chemical denaturation experiments using urea or GdmCl, respectively, (*Figure 6—figure supplement 5A, B*), which clearly show that the transition during the titration from 0 to ~3 M urea/~1 M GdmCl is accompanied by a loss in tertiary structure. The possibility that the resulting conformation corresponds to an equilibrium molten globule is, however, excluded due to its inability to bind 8-anilino-1-naphthalenesulfonic acid (ANS, *Figure 6—figure supplement 5C*).

In order to identify signals corresponding to the minor species in the HSQC spectra of *Ec*RfaH-KOW, we started with the spectrum of the urea-unfolded protein (8 M urea, purple spectra in *Figure 6B and C*). Most of the corresponding signals shifted linearly with decreasing urea concentration and also lost intensity at urea concentrations <3 M (e.g. signal of S139 in *Figure 6B and C*). At 0 M urea, finally, only a set of weak signals remained, which we identified as signals of the minor species as these match the chemical shifts of the minor species identified in the CEST experiments (compare red and gray bars in *Figure 6A*, top/bottom panels). Based on the linear transition between the positions of the (urea) unfolded state toward the positions of the minor species signals, we conclude that addition of urea shifts the minor species' population toward the completely unfolded state. Although we cannot assess if the minor species samples the completely unfolded state in the absence of any denaturant, the increased $^{15}$N $R_2$ values indicate additional exchange processes of the minor species on the μs-ms time scale (*Figure 6A*, middle panel). Thus, we hypothesize that the minor species can be described as an ensemble of exchanging sub-states, some corresponding to the completely unfolded state 'U' and some exhibiting residual helical structure, hereby referred to as α-helical unfolding intermediate 'Uα' with the minor species observed in the CEST experiments being the average population under native conditions.

If this is true, the urea-induced chemical shift perturbations experienced by the minor species signals in the [$^1$H, $^{15}$N]-HSQCs can be explained by a combination of two effects: (i) change of the chemical environment of the spins due to the presence of urea, which particularly affects δ $^1$H (see e.g. signal of T157 in *Figure 6B*), and (ii) change in the relative populations of the minor species' sub-states toward the unfolded state, which mainly affects $\Delta\delta^{15}$N. Since the Hα/Cα chemical shifts are relatively independent of the solvent conditions, their perturbations in the urea denaturation series (*Figure 6C*) even better reflect the change in the ratio of the minor species' sub-states. The shifting of the minor species' peaks in *Figure 6C* is completed at ≈7 M urea, implying that the second transition in the far-UV CD-based unfolding experiment (*Figure 3F*) corresponds to the denaturation of Uα. Interestingly, the $R_2$ values of residues in region 1 are more than twice as high as those of residues in region 2 (*Figure 6A*) and, in the [$^1$H, $^{15}$N]-HSQC-based denaturation experiment (*Figure 6B*), the minor species' signals of residues in region 1 do not shift in a linear manner as it is typical for two exchanging states. Instead, they show a curved transition that is 'kinked' at ≈ 2 M urea (see e.g. T131), implying a more complex unfolding process and thus structural heterogeneity of this region. Although our experiments do not allow a precise structural characterization of all states of the minor species, it may be described as an ensemble of largely unfolded, interconverting structures with states U and Uα constituting the extrema.

Due to the fast chemical exchange between the *Ec*RfaH-KOW's U and Uα states during the chemical denaturation, their relative populations in a certain titration step are encoded in the chemical shift of the minor species signal, whereas the volume of the minor species peak is proportional to the sum of the populations of both states (assuming similar transverse relaxation rates for the species). The chemical shifts of Cα/Hα groups depend to a much lower extent on the urea concentration in the sample than the chemical shifts of amide groups and therefore they provide better measures for the exchange between U and Uα. To first quantify the decay of the all-β conformation and the increase of the minor species during the urea denaturation, we analyzed the peak intensity of both species exemplarily for residue S139 in the [$^1$H, $^{13}$C]-ctHSQC-based titration (*Figure 6C* and *Figure 6—figure supplement 6A*). The resulting $\Delta G$ value of ≈ 7 kJ/mol between the energy levels of major and minor species agrees well with the results from the CEST experiment (7 kJ/mol). Additionally, the $m$ value of 3.4 kJ/(mol M) is very similar to the $m$ values obtained for the other KOW domains by CD spectroscopy

(*Table 3*), indicating that the minor species is indeed close to a completely unfolded state with a small buried surface area.

The complete denaturation of the minor species, that is, the transition of Uα to a fully unfolded state U, can be followed in the [$^1$H, $^{13}$C]-ctHSQC-based denaturation experiment by analyzing the change of the minor species's chemical shifts from the positions in the absence of urea toward those of the completely unfolded state. For example, the Hα/Cα correlation peaks of residues A137, S139, or M140, which are located in region 2, clearly shift from regions typical for α-helical structures (upfield $^1$H, downfield $^{13}$C relative to random coil values) to positions corresponding to an unstructured conformation (downfield $^1$H, upfield $^{13}$C), and finally they localize next to the signals of the Ala, Ser, or Met residues that do not reside in regions with residual helical structure (*Figure 6C*). Plotting the changes of the $^{13}$Cα chemical shifts of A137, S139, and M140 versus the urea concentration (*Figure 6—figure supplement 6B*) results in curves that resemble the second half of an unfolding transition (Uα ⇌ U) and approach the baseline of the fully unfolded state at ≈6 M urea. The absence of a baseline for Uα precludes a quantitative analysis, but it indicates that the transition mid-point of the curve is probably close to or below 0 M urea. In summary, the data of the NMR-based denaturation experiments (i) strongly support our hypothesis that the minor species identified in the CEST experiments is an ensemble of fast interconverting, mostly unfolded structures with U and Uα being the extrema and (ii) suggest that the minor species might be an important intermediate during the refolding process.

## Discussion

### Fold-switching is conserved among RfaH proteins

Genes coding for RfaH orthologs can be found in many bacterial pathogens, including *Salmonella*, *Klebsiella*, *Vibrio*, and *Yersinia* spp. (*Carter et al., 2004*). Despite their divergent evolution, RfaH proteins seem to have a conserved mechanism of action (*Carter et al., 2004*). To date, only *Ec*RfaH was structurally characterized in detail, revealing that this protein has unique structural features classifying it as transformer protein (*Belogurov et al., 2007*; *Burmann et al., 2012*; *Zuber et al., 2019*). Here, we show that *Vc*RfaH, an evolutionary quite divergent representative sharing 35.8% sequence identity with *Ec*RfaH, exhibits very similar structural properties, that is, *Vc*RfaH-KOW, like *Ec*RfaH-KOW, folds as α-hairpin in the full-length protein, but adopts a NusG-type β-barrel conformation in its isolated form (*Figure 1*). Interestingly, in *Vc*RfaH helix α$_3$* is 1.5 turns longer as compared to *Ec*RfaH and *Vc*RfaH has a disulfide bridge connecting strand β$_3$* and helix α$_3$*, stabilizing this helix. These two features imply a stabilization of the domain interface and thus an increased affinity between the domains as compared to *Ec*RfaH. This might also explain the increased stability of the isolated *Vc*RfaH-KOW domain (≈14 kJ/mol), which compensates the higher energy gain of the domain interaction. Further, the increased stability of the *Vc*RfaH-KOW domain may be the cause for the sigmoid-shaped CD-based chemical denaturation curves, in agreement with an apparent two-state unfolding process: global unfolding of the folded state occurs at higher denaturant concentrations, where potential partly structured folding intermediates are already largely destabilized and therefore escape detection. This conclusion is supported by the Trp fluorescence-based denaturation data (*Figure 3—figure supplement 1*), suggesting that the change in the CD signal is almost exclusively caused by the decay of the β-barrel conformation and that the contribution of Uα to the change of the CD signal is negligible. Nevertheless, we conclude that *Vc*RfaH may be regulated by fold-switching just like *Ec*RfaH, and that this metamorphic behavior is conserved in the class of RfaH proteins and may even be found in other NusG paralogs, in agreement with a recent study that predicts that nearly 25% of bacterial NusG proteins might perform α ↔ β transitions similar to *Ec*RfaH (*Porter et al., 2022*).

### Model for the structural plasticity of RfaH

*Ec*RfaH switches the conformation and function of its KOW domain in a reversible manner to achieve a tight control of gene expression (*Zuber et al., 2019*). In free *Ec*RfaH, the α-helical hairpin conformation is the preferred state of *Ec*RfaH-KOW, whereas domain separation or isolation of *Ec*RfaH-KOW fosters population of the all-β state in solution (*Burmann et al., 2012*), suggesting that the all-α conformation is intrinsically unstable, but becomes the thermodynamic minimum in free *Ec*RfaH due to interaction with *Ec*RfaH-NGN.

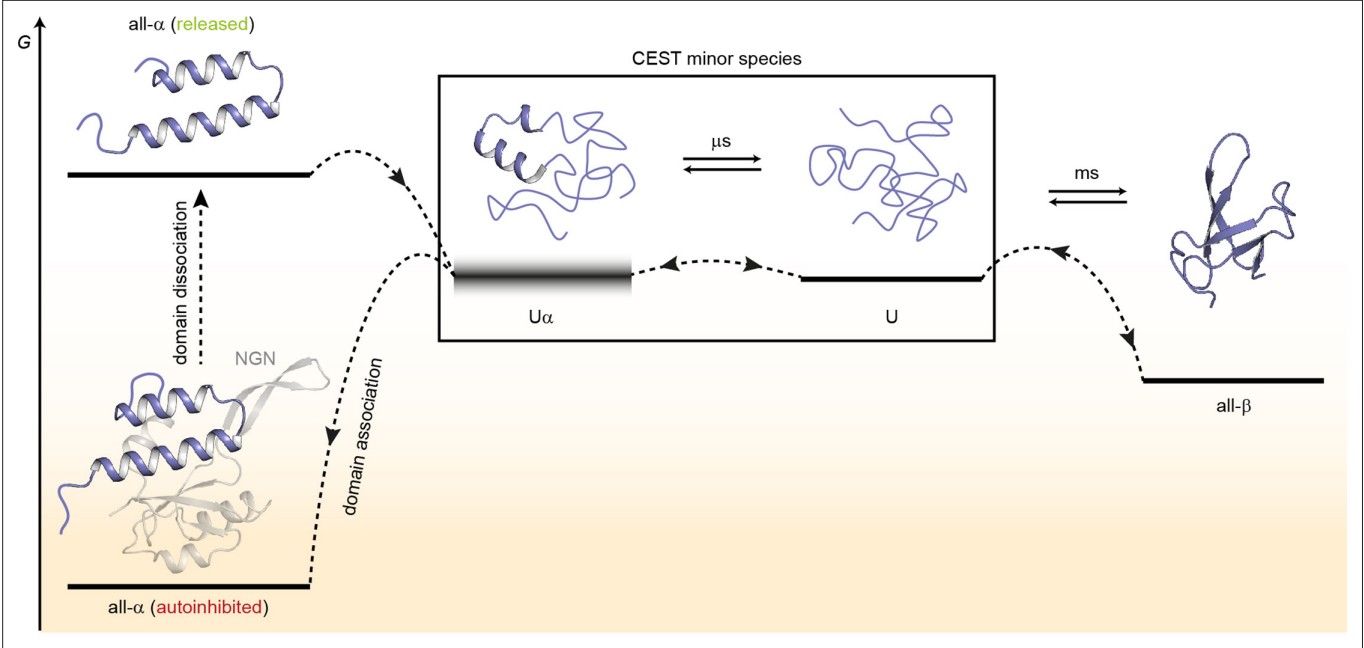

**Figure 7.** Model for the conformational plasticity of *Ec*RfaH-KOW. Qualitative Gibbs free energy level diagram and associated structures for the all-α to all-β transition of *Ec*RfaH-KOW and vice versa. In its ground state, that is, the autoinhibited conformation, the energy of the all-α conformation of *Ec*RfaH-KOW is strongly lowered by the extensive inter-domain contacts with the *Ec*RfaH-NGN domain. Upon recruitment, the domains dissociate, the helical structure of the released KOW domain becomes destabilized in isolation, and rapidly decays toward an ensemble of mainly unfolded sub-states that interconvert on the μs time scale. Some of the sub-states correspond to the completely unfolded state (**U**) whereas others retain some residual (α-) helical elements (Uα). The scheme displays exemplary structures of these sub-states. Due to their fast structural interconversion, U and Uα may be grouped into a single macro-state/ensemble (as is the case during the chemical exchange saturation transfer [CEST] experiments) that exhibits helical structures for a limited amount of time and is otherwise unfolded. Uα is either marginally stable or even unstable (therefore, its energy level is blurred). The disordered conformation then allows for easy and rapid refolding to the all-β conformation. Due to their low thermodynamic stability, or even instability of all-β and Uα, respectively, the last two steps are reversible, that is, the all-α state can be rapidly regained when the *Ec*RfaH-NGN domain becomes available for re-association after transcription termination.

Interestingly, our thermodynamic analysis (*Figures 2 and 3*) of the isolated *Ec*RfaH-KOW domain reveals that, although the all-β conformation is the preferred state in isolation, it is only marginally stable, and it is in rapid equilibrium with an 'unfolded' state, which is populated to a significant extent, even under physiological conditions. The 'unfolded' state is a mixture of random-coil-type unfolded species U and species Uα containing two helical regions.

Based on our results, we suggest a model for the structural transitions of *Ec*RfaH-KOW (*Figure 7*).

In the autoinhibited state the all-α conformation of *Ec*RfaH-KOW corresponds to the minimum of the Gibbs free energy as it is stabilized by contacts to the *Ec*RfaH-NGN. During recruitment of *Ec*RfaH to an *ops*-paused elongation complex, the *Ec*RfaH-NGN:KOW interface is destabilized (most probably via an encounter complex), the domains dissociate and *Ec*RfaH-NGN is sequestered to RNAP (*Zuber et al., 2019*). The freed all-α *Ec*RfaH-KOW is not stable as *G* increases due to the loss of *Ec*RfaH-NGN contacts. Consequently, *Ec*RfaH-KOW unfolds, resulting in an ensemble of rapidly interconverting sub-states. Some of these sub-states still contain two residual α-helical regions (intermediate Uα) that correspond to the tip of the α-hairpin in the all-α state, in agreement with hydrogen/deuterium exchange data, which indicate that the hairpin tip is the most stable part of the all-α conformation (*Galaz-Davison et al., 2020*). Other sub-states represent the completely unfolded protein, which then rapidly refolds into the all-β form. Upon transcription termination *Ec*RfaH is released, and the process is reversed with unfolding of the β-barrel starting, most probably, by detaching β1 and β4/β5 from the central strands as the corresponding H-bonds are the least stable ones (*Figure 4*). The U state is in equilibrium with Uα, where two α-helical regions that will later constitute the α-hairpin tip are formed

transiently and may thus serve both as the nucleation point for the completion of the all-α structure and as starting point for recognition of its cognate binding site on the NGN. This mechanism ensures rapid re-autoinhibition and prevents aggregation of *Ec*RfaH. Although we did not analyze *Vc*RfaH as extensively as *Ec*RfaH, our results suggest that the *Vc*RfaH-KOW domain most likely employs a similar mechanism for its structural transformation, indicating that the presented model is a general scheme for RfaH proteins.

In support of our model, all computational studies on *Ec*RfaH found that the all-α conformation is stable only when in contact with the NGN. Modification of the strength of the *Ec*RfaH-NGN:KOW interface (*Ramírez-Sarmiento et al., 2015*) or deletion of the linker (*Xun et al., 2016*) destabilizes the all-α fold and ultimately drives *Ec*RfaH-KOW into the β-barrel state. Moreover, the β-barrel fold is stable and corresponds to or is close to the energy minimum of the energy landscape of *Ec*RfaH-KOW, whereas the all-α fold rapidly unfolds and has a higher *G* value than the all-β state (*Balasco et al., 2015*; *Bernhardt and Hansmann, 2018*; *Gc et al., 2014*; *Joseph et al., 2019*; *Li et al., 2014*; *Xiong and Liu, 2015*). Apart from these general concepts, most studies differ in several key points, such as the extent to which the all-α state is populated in the isolated *Ec*RfaH-KOW, or the precise folding pathway from all-α to all-β. Strikingly, a recent bioinformatical study very nicely mirrors our data as the authors also observed a significant portion of transiently formed helical structure within the unfolded state ensemble in their simulations (*Seifi and Wallin, 2021*).

## Requirements for fold-switching proteins

Previous work on designed and naturally occurring fold-switching proteins has identified several specific properties that make fold-switching proteins distinct from others (*Bryan and Orban, 2010*; *Porter and Looger, 2018*). In this study, we show that RfaH meets all these requirements and is thus a showcase example for fold-switching proteins:

1. Reduced thermodynamic stability (*Bryan and Orban, 2010*). A diminished stability is both the result of and key to the function of fold-switching proteins. As the fold-switching sequence must be compatible with both adopted topologies, it can only be optimized to a certain extent to stabilize one specific fold, ensuring that both conformations can be interconverted and that the structure is not 'trapped' in one state. This is reflected by a dual-funneled energy landscape with two main minima, which are, however, not as deep as the global minimum of a stable protein. Our comprehensive thermodynamic analysis (*Figures 2 and 3*) reveals that the all-β fold of both RfaH-KOWs is indeed less stable than the bacterial and archaeal NusG/Spt5-KOW domains. As general transcription factors, NusG/Spt5 proteins do not require an as-sophisticated regulation as RfaH (*Artsimovitch and Knauer, 2019*) (for hSpt5-KOW5 see below) and thus benefit from a stable structure to carry out their function. The difference in thermodynamic stability is especially striking for *Ec*NusG-KOW and *Ec*RfaH-KOW as both belong to the same class of transcriptional regulators, originate from the same organism, and share a sequence identity of 35.8% (43.6% for the full-length proteins), yet underly completely different regulatory mechanisms that, in turn, strongly depend on the difference of this thermodynamic parameter. As a result, *Ec*RfaH is tightly regulated by autoinhibition coupled to the conformational transformation of a whole domain and controls just a small set of specific genes whereas *Ec*NusG is a stable, monomorphic protein involved in the transcription of most host genes.

2. Generation of new binding surfaces (*Bryan and Orban, 2010*). The regulation of conformational transitions in fold-switching proteins is achieved by energetically stabilizing one of the two conformations in response to a molecular trigger, resulting in a far more dynamic energy landscape than that of well-folded, monomorphic proteins as the energy level of a particular conformation strongly depends on the environment. This context-dependent stabilization of one state is possible because the two different folds exhibit different surface topologies, each allowing distinct interactions. The ability to selectively hide/expose 'latent' binding sites within different folds is also the most important function of fold-switching in general, as it enables a level of control that cannot be achieved by other mechanisms. In RfaH, autoinhibition is coupled to a conformational switch. In the autoinhibited state the all-α KOW interacts with the RfaH-NGN to prevent off-target recruitment and interference with NusG (*Belogurov et al., 2009*), whereas the refolded state allows simultaneous binding of RfaH to RNAP via RfaH-NGN and to the ribosome via all-β RfaH-KOW to activate translation (*Kang et al., 2018*; *Zuber et al., 2019*).

3. Unfolded regions in one of the two states (*Bryan and Orban, 2010*). In RfaH-KOW, the central β-strands $\beta_2$, $\beta_3$, and $\beta_4$ transform into two α-helices during the all-β-to-all-α transition and vice versa (*Figure 1B*). However, the all-α KOW domain contains unstructured N- and C-termini, whereas the corresponding regions form β-strands $\beta_1$ and $\beta_4/\beta_5$ in the all-β conformation (*Figure 1B* and *Figure 4C*). These disordered parts provide an entropic stabilization of the respective state as they do not adopt a defined structure and the corresponding β-strands are less stably bound to the rest of the β-barrel than in the stable KOW domains (*Figure 4*). A bioinformatic study indicated that these regions of the CTD additionally stabilize the NGN:KOW interface by forming transient, IDP-like interactions (*Xun et al., 2016*). We show that the structural interconversion between the two RfaH states proceeds via a chiefly unfolded intermediate and we propose that the disordered segments may help to facilitate and/or initiate this transition, similar to the mechanism suggested for the human chemokine XCL1 (lymphotactin) (*Tyler et al., 2011*). Finally, disordered regions in one state have the advantage that they can be evolutionary optimized to selectively stabilize one of the two states of a fold-switch pair, whereas there is no need to fit a defined three-dimensional structure in the other state. This is reflected by the secondary structure predictions of both *Ec*RfaH-KOW and *Vc*RfaH-KOW, which suggest propensities for both helical structures and β-strands in the central region that indeed interconvert between α-helices and β-strands, whereas only β-strands are predicted for the termini (*Figure 6—figure supplement 2*).

4. Divergence in predicted and observed secondary structure (*Porter and Looger, 2018*). Secondary structure predictions show that both *Vc*RfaH-KOW and *Ec*RfaH-KOW contain stretches with high propensity for both β-strands and α-helical structures, whereas NusG/Spt5-KOW domains are predicted to adopt four to five β-strands (*Figure 6—figure supplement 2*). Three-dimensional structures of the KOW domains of this study confirm that the NusG/Spt5-KOW domains are indeed β-barrels, whereas the fold of the RfaH-KOW domains depends on the context (*Figure 1* and *Figure 1—figure supplement 2*). Interestingly, this tendency is also visible in the isolated KOW domain as the disordered regions in the all-α fold correspond to the β-strands that are less stable in the RfaH-KOWs as compared to NusG-KOWs, whereas the helical propensity is reflected in the structure of Uα. However, one should keep in mind that secondary structure predictions strongly depend on the underlying algorithms, as can be seen for *Mt*NusG-KOW (*Figure 6—figure supplement 2*).

5. Cooperatively folding units (*Porter and Looger, 2018*). The folding cooperativity of *Ec*RfaH-KOW depends on the presence of the *Ec*RfaH-NGN, that is, in the absence of *Ec*RfaH-NGN *Ec*RfaH-KOW folds cooperatively on its own. However, the cooperativity is generally rather low and the activation barrier separating the 'unfolded' and the folded states is small, allowing fast transitions.

## Fold-switching is a highly efficient principle of regulation with a steadily increasing importance

To date, about six fold-switching proteins have been studied in detail (summarized in *Dishman and Volkman, 2018*; *Lella and Mahalakshmi, 2017*; *Zamora-Carreras et al., 2020*), but estimates suggest that up to 4% of the proteins in the PDB may have the ability to switch folds (*Porter and Looger, 2018*). Our study demonstrates which molecular mechanisms confer RfaH its structural plasticity that allows operon-specific regulation without competing with its monomorphic paralog NusG/Spt5. In line with our findings, a recent study on XCL1, another model system for fold-switching proteins, identified very similar principles for the evolution and design of fold-switching proteins (*Dishman et al., 2021*).

## Importance of a chiefly unfolded state in protein fold-switching

In summary, our results highlight two key features in protein fold-switching: decreased thermodynamic stability and defined local structures in 'unfolded' intermediates. Diminished stability is often thought to be detrimental for proteins as it favors non-native contacts and promotes aggregation. However, it is essential to confer fold-switching proteins their conformational plasticity, and, as all transitions from and to the unfolded states are very fast, and the population of these states is rather low, fold-switchers can evade aggregation. Further, the capability of the 'unfolded' state to harbor residual defined structures, for example, α-helices, allows to pre-encode a second conformation that could be readily adopted upon a molecular signal.

# Materials and methods

## Key resources table

| Reagent type (species) or resource | Designation | Source or reference | Identifiers | Additional information |
|---|---|---|---|---|
| strain, strain background | *Escherichia coli* BL21(DE3) | Novagen | N/A | |
| recombinant DNA reagent | List of recombinant plasmids used | **Table 7** | | |
| Sequence-based reagent | List of primers used | **Table 6**, Biolegio | PCR primers | |
| peptide, recombinant protein | *V. cholerae* RfaH | This work | | See Materials and methods, section 'Production of recombinant proteins' |
| peptide, recombinant protein | *V. cholerae* RfaH-KOW | This work | | See Materials and methods, section 'Production of recombinant proteins' |
| peptide, recombinant protein | *E. coli* RfaH-KOW | **Burmann et al., 2012** doi:10.1016/j.cell.2012.05.042 | | |
| peptide, recombinant protein | *E. coli* NusG-KOW | **Burmann et al., 2010** doi: 10.1126/science.1184953 | | |
| peptide, recombinant protein | *M. tuberculosis* NusG-KOW | **Strauß et al., 2016** doi: 10.1080/07391102.2015.1031700 | | |
| peptide, recombinant protein | *M. janaschii* Spt5-KOW | This work | | See Materials and methods, section 'Production of recombinant proteins' |
| peptide, recombinant protein | Human Spt5-KOW5 (G699-G754) | This work | | See Materials and methods, section 'Production of recombinant proteins' |
| commercial assay or kit | QIAquick Gel Extraction Kit | Qiagen | Cat#: 28706 | |
| commercial assay or kit | QIAprep Spin Miniprep Kit | Qiagen | Cat#: 27106 | |
| chemical compound, drug | $(^{15}NH)_4SO_4$ | Sigma/Merck KGaA | Cat#: CS01-185_148 | |
| chemical compound, drug | $D_2O$ | Euriso-Top GmbH | Cat#: 7789-20-0 | |
| chemical compound, drug | $^{13}C$-D-glucose | Euriso-Top GmbH | Cat#: CLM-1396–10 | |
| chemical compound, drug | Urea | Carl Roth GmbH & Co. KG | Cat#: 2317.1 | |
| chemical compound, drug | GdmCl | Carl Roth GmbH & Co. KG | Cat#: 0037.1 | |
| chemical compound, drug | DSS | Sigma | Cat#: T-8636 | |
| chemical compound, drug | ANS | Sigma/Merck KGaA | Cat#: 10417–5G-F | |
| software, algorithm | Fit-o-Mat v0.752 | **Möglich, 2018** doi: 10.1021/acs.jchemed.8b00649 | | |
| software, algorithm | PyMol v. 1.7 | The PyMOL Molecular Graphics System, Schrödinger, LLC | https://pymol.org/2/ | |
| software, algorithm | NMRViewJ | One Moon Scientific, Inc | http://www.onemoonscientific.com/nmrviewj | |

*Continued on next page*

*Continued*

| Reagent type (species) or resource | Designation | Source or reference | Identifiers | Additional information |
|---|---|---|---|---|
| software, algorithm | ChemEx v. 0.6.1 | ***Vallurupalli et al., 2012*** doi:10.1021/ja3001419 | https://github.com/ gbouvignies/ ChemEx | |
| other | Quartz cuvette for CD spectroscopy, 1 mm | Hellma GmbH & Co. KG | | See Materials and methods, section 'CD spectroscopy' |
| other | Quartz cuvette for CD spectroscopy, 2 mm | Hellma GmbH & Co. KG | | See Materials and methods, section 'CD spectroscopy' |
| other | Quartz cuvette for fluorescence spectroscopy, 1 cm | Hellma GmbH & Co. KG | | See Materials and methods, section 'Fluorescence spectroscopy' |

## Cloning

The *Vc*RfaH expression vector pVS13 (*V. cholerae rfaH* from pHC301 (*Carter et al., 2004*) in plasmid pTYB1 [NEB]) was a gift from I Artsimovitch, The Ohio State University, Columbus, OH. The C-terminal *Vc*RfaH residue, Thr165, is substituted by an Ala to ensure efficient cleavage of the resulting chitin binding domain (CBD) intein fusion protein (see below). Expression plasmids for *Vc*RfaH-KOW (residues E103-T165), hSpt5-KOW5 (residues G699-G754), and *Mj*Spt5-KOW (residues K83-D147) were created by cloning of the corresponding gene regions into vector pETGb1a (G Stier, EMBL, Heidelberg, Germany) via *Nco*I and *BamH*I (*Vc*RfaH-KOW), or *Nco*I and *EcoR*I (hSpt5-KOW5 and *Mj*Spt5-KOW) restriction sites, respectively. Templates for PCR amplification were plasmids pHC301 (*Carter et al., 2004*) for *Vc*RfaH-KOW, pOTB7_huSUPT5H (*Zuber et al., 2018*) for hSpt5-KOW5, and pGEX-2TK_*Mj*Spt5-KOW (*Hirtreiter et al., 2010*); kindly provided by F Werner, University College London, UK for *Mj*Spt5-KOW. The primers used for cloning are listed in *Table 6*. All plasmids used in this study are listed in *Table 7*.

## Production of recombinant proteins

*Vc*RfaH was obtained from a CBD intein fusion protein encoded in plasmid pVS13, with expression conditions and purification strategy as described for *E. coli* RfaH (*Vassylyeva et al., 2006*). *Ec*NusG-KOW and *Mt*NusG-KOW were produced as previously described (*Burmann et al., 2010*; *Strauß et al., 2016*). *Mj*Spt5-KOW, hSpt5-KOW5, *Ec*RfaH-KOW, and *Vc*RfaH-KOW were obtained from Gb1 fusions with expression and purification conditions similar to that of *Ec*RfaH-KOW (*Burmann et al., 2012*).

The quality of all recombinantly produced proteins was ensured according to the guidelines established by ARBRE-MOBIEU and P4EU (https://arbre-mobieu.eu/guidelines-on-protein-quality-control/) (*de Marco et al., 2021*). In brief, purity was checked by sodium dodecyl sulfate polyacrylamide gel electrophoresis, the absence of nucleic acids by UV spectroscopy, the identity by mass spectrometry and/or NMR spectroscopy, the folding state by CD and/or NMR spectroscopy, and the absence of aggregation by analytical gel filtration or dynamic light scattering.

**Table 6.** Primers used for cloning.

| Primer | Sequence (5' → 3') |
|---|---|
| Fw-*Vc*RfaH-KOW | CAT GCC ATG GGA GAG CAA TTG AAG CAT GCC AC |
| Rv-*Vc*RfaH-KOW | CGC GGA TCC TTA GGT GAC TTC CCA ATC GG |
| Fw-hSpt5-KOW5 | CAT GCC ATG GGC CGG AGG GAC AAC GAA CTC ATC GG |
| Rv-hSpt5-KOW5 | TAG AAT TCT CAG CCC ACC GTG GTG AGC CGC TG |
| Fw-*Mj*Spt5-KOW | AT GCC ATG GGT AAG AAA ATC ATT GAA AAT ATT GAG AAA GG |
| Rv-*Mj*Spt5-KOW | CGG AAT TCT TAA TCT TTA TGC TTT GAA ACT ATT TTA AC |

**Table 7.** Plasmids.

| Plasmid | Description | Source |
|---|---|---|
| pVS13 | *rfaH* from *V. cholera* in pTYB1 | I Artsimovitch |
| pHC301 | *rfaH* from *V. cholera* in pIA238 (a pET28 derivative) **Artsimovitch and Landick, 2002** | **Carter et al., 2004** |
| pETGb1a-*Vc*RfaH-KOW | *rfaH*[103-165] from *V. cholera* in pETGb1a | This work |
| pETGb1a-hSpt5-KOW5 | human *spt5*[699-754] in pETGb1a | This work |
| pETGb1a-*Mj*Spt5-KOW | *spt5*[583-147] from *M. janaschii* in pETGb1a | This work |
| pOTB7_huSUPT5H | cDNA plasmid containing human *spt5* | **Zuber et al., 2018** |
| pGEX-2TK_*Mj*Spt5-KOW | *spt5*[583-147] from *M. janaschii* in pGEX-2TK | **Hirtreiter et al., 2010** |
| pETGb1a-*Ec*NusG-KOW | *nusG*[123-181] from *E. coli* in pETGb1a | **Burmann et al., 2010** |
| pET101d-*Mt*NusG-KOW | *nusG*[178-238] from *M. tuberculosis* in pET101d | **Strauß et al., 2016** |
| pETGb1a-*Ec*RfaH-KOW | *rfaH*[101-162] from *E. coli* in pETGb1a | **Burmann et al., 2012** |

## Isotopic labeling of proteins

For the production of $^{15}$N- and $^{15}$N, $^{13}$C-labeled proteins, *E. coli* cells were grown in M9 medium (**Green et al., 2012**; **Meyer and Schlegel, 1983**) containing ($^{15}$NH$_4$)$_2$SO$_4$ (Sigma/Merck KGaA, Darmstadt, Germany) or ($^{15}$NH$_4$)$_2$SO$_4$ and $^{13}$C-D-glucose (Euriso-Top GmbH, Saarbrücken, Germany), respectively, as sole nitrogen or carbon sources. Deuteration was achieved by accustoming cells to M9 medium prepared with increasing concentrations of D$_2$O (0%, 50%, 100% (v/v); Euriso-Top GmbH, Saarbrücken, Germany). Expression and purification protocols were identical to those of the unlabeled proteins.

## NMR spectroscopy

NMR experiments were conducted at Bruker Avance 600, Avance 700, Ascend Aeon 900, and Ascend Aeon 1000 spectrometers, each equipped with room temperature (Avance 600) or cryogenically cooled, inverse $^1$H, $^{13}$C, $^{15}$N triple resonance probes (all other spectrometers). All measurements were conducted in 5 mm tubes with a sample volume of 550 µl at 25°C, if not stated otherwise. NMR data was processed using in-house software and analyzed using NMRViewJ (OneMoon Scientific).

Backbone resonance assignments for *Vc*RfaH, *Vc*RfaH-KOW, hSpt5-KOW5, *Mj*Spt5-KOW, and urea-unfolded *Ec*RfaH-KOW were obtained using standard band-selective excitation short transient (**Lescop et al., 2007**; **Schanda et al., 2006**) transverse relaxation optimized spectroscopy (TROSY)-based triple resonance experiments (**Pervushin et al., 1997**; **Salzmann et al., 1998**). Additionally, carbon-detected CACO, CAN, and NCO experiments (**Bermel et al., 2005**) were recorded for *Vc*RfaH-KOW. Side chain assignments for *Vc*RfaH-KOW were obtained from CCH- and H(C)CH-TOCSY, HBHA(CO)NH, C(CO)NH, aromatic [$^1$H, $^{13}$C]-HSQC, and $^{13}$C-edited aromatic nuclear overhauser enhancement spectroscopy (NOESY) experiments (**Sattler et al., 1999**). Three-dimensional assignment and NOESY experiments were acquired using a non-uniform sampling scheme with a sparsity of 25–50%. Spectra were subsequently reconstructed with in-house written software using the iterative soft thresholding algorithm (**Hyberts et al., 2012**). The *Ec*RfaH-KOW, *Vc*RfaH-KOW, hSpt5-KOW5, and *Mj*Spt5-KOW samples contained 0.5–1 mM [$^{15}$N, $^{13}$C]-labeled protein in 20 mM Na-phosphate (pH 6.5), 100 mM NaCl, 1 mM ethylenediaminetetraacetic acid (EDTA), 10% (v/v) D$_2$O. The *Ec*RfaH-KOW sample further contained 6 M urea. Due to limited sample stability and poor quality of the initial spectra, *Vc*RfaH (0.3 mM) was [$^2$H, $^{15}$N, $^{13}$C]-labeled and in an optimized buffer (25 mM Bis-Tris-Propane [pH 6.5], 25 mM Na-Tartrate, 50 mM NaCl, 10% (v/v) D$_2$O) and the measurements were conducted at 20°C. The Cα and CO secondary chemical shift for *Vc*RfaH was calculated as difference between the observed chemical shift and the predicted random coil value (**Wishart and Sykes, 1994**) using a deuterium correction as given in **Venters et al., 1996**. Chemical shift assignments for *Ec*NusG-KOW, *Mt*NusG-KOW, and native *Ec*RfaH-KOW were taken from previous studies (**Burmann et al., 2012**; **Mooney et al., 2009**; **Strauß et al., 2016**). The random coil chemical shifts for characterization of the minor species in case of *Vc*RfaH-KOW and hSpt5-KOW were calculated using the Poulsen IDP/IUP random coil chemical shifts calculator tool (https://spin.niddk.nih.gov/bax/nmrserver/Poulsen_rc_CS/).

Distance restraints for the structure calculation of *Vc*RfaH-KOW were obtained from standard $^{13}$C- and $^{15}$N-edited 3D NOESY experiments (*Sattler et al., 1999*) with mixing times of 120 ms. NOESY cross-signals were classified according to their intensities and converted to distance restraints with upper limits of 3 Å (strong), 4 Å (medium), 5 Å (weak), and 6 Å (very weak), respectively. Hydrogen bonds were identified from corresponding experiments (see below). Psi/Phi angle restraints were obtained from the geometry dependence of the backbone chemical shifts using TALOS (*Cornilescu et al., 1999*). The structure calculation was performed with XPLOR-NIH version 2.1.2 using a three-stage simulated annealing protocol with floating assignment of prochiral groups including a conformational database potential (*Schwieters et al., 2003*). Structures were analyzed with XPLOR-NIH and PROCHECK-NMR (*Laskowski et al., 1996*).

$^{15}$N-based CEST experiments were conducted according to *Vallurupalli et al., 2012*. All samples contained ≈0.7–1 mM $^{15}$N-labeled protein. For initial CEST experiments, the domains were in 20 mM HEPES (pH 7.5), 100 mM NaCl, 10% (v/v) D$_2$O, and a single CEST B$_1$ field ($\nu_1$=18–25 Hz) during an exchange period of 500 ms was employed. Proteins showing an exchange peak in their CEST profiles were further studied in 20 mM Na-phosphate (pH 6.5), 100 mM NaCl, 1 mM EDTA, 10% (v/v) D$_2$O to decrease amide proton-H$_2$O exchange. CEST experiments were then recorded using two different B$_1$ fields ($\nu_1$=13 Hz/26 Hz) and an exchange period of 500 ms. The B$_1$ frequencies were calibrated using a 1D approach on an isolated signal (*Guenneugues et al., 1999*). The CEST traces obtained at 13/26 Hz were fitted simultaneously according to a two-state exchange model using ChemEx (version 0.6.1, *Vallurupalli et al., 2012*). Due to the monodisperse distribution of the resulting $k_{ex}$/$p_B$ values (*Table 5*), the CEST traces were then fitted globally, yielding a global $k_{ex}$ and $p_B$ value. Only those CEST profiles were included in the global fit that showed a $\Delta\omega$>1 ppm. $^{13}$Cα-CEST experiments were recorded on [$^{15}$N, $^{13}$C]-labeled protein samples using a [$^{1}$H, $^{13}$C] constant-time (ct) HSQC-based approach (*Bouvignies et al., 2014*). To maximize the number of analyzable signals, the proteins were in 20 mM Na-phosphate (pH 6.5), 100 mM NaCl, 1 mM EDTA, 99.9% (v/v) D$_2$O (pH uncorrected for D$_2$O). In this case, the chemical shift was referenced via 0.5 mM internal DSS. The experiment was performed at a single B$_1$ field strength (25 Hz) at an exchange period of 500 ms. The CEST traces obtained for [$^{1}$H, $^{13}$C]-*Ec*RfaH-KOW were fitted with ChemEx.

NMR-based chemical denaturation experiments of the KOW domains were done by recording [$^{1}$H, $^{15}$N]-HSQC and [$^{1}$H, $^{13}$C]-ctHSQC spectra of 80 µM [$^{15}$N, $^{13}$C]-*Ec*RfaH-KOW in 20 mM Na-phosphate (pH 6.5), 100 mM NaCl, 1 mM EDTA, 10% (v/v) D$_2$O buffer containing 0–8 M urea. The chemical shifts were referenced to 0.5 mM internal DSS.

For the NMR-based refolding experiment of *Vc*RfaH under reducing conditions a [$^{1}$H, $^{15}$N]-HSQC spectrum of $^{15}$N-*Vc*RfaH in refolding buffer (50 mM Na-phosphate [pH 6.5], 50 mM NaCl, 2 mM DTT) was recorded before the protein was incubated in refolding buffer containing 8 M urea for 24 hr. Having recorded another [$^{1}$H, $^{15}$N]-HSQC spectrum urea was removed by stepwise dialysis against 4 l of refolding buffer containing 4, 2, 1, 0.5, and 0 M urea, respectively (2–4 hr for the first four steps and over-night for the last step). Finally, a [$^{1}$H, $^{15}$N]-HSQC spectrum of the refolded protein was recorded.

Hydrogen bonds were identified from 2D or 3D long-range TROSY-based HNCO experiments as previously described (*Cordier et al., 2008*). All samples contained [$^{15}$N, $^{13}$C]-labeled proteins at 0.7–1 mM in 20 mM Na-phosphate (pH 6.5), 100 mM NaCl, 1 mM EDTA, 10% (v/v) D$_2$O.

## CD spectroscopy

CD data were collected at a Jasco J-1100 spectrometer (Jasco Deutschland GmbH, Pfungstadt, Germany), using quartz cuvettes (Hellma GmbH & Co. KG, Müllheim, Germany). CD spectra were normalized (*Equation 1*) to obtain the mean residue-weighted ellipticity ($\Theta_{MRW}$):

$$\Theta_{MRW} = \frac{100 \cdot \theta}{N \cdot c \cdot d} \qquad (1)$$

$\theta$ is the ellipticity in mdeg, $N$ the number of amino acids, $c$ the protein concentration in mM, and $d$ the pathlength of the cuvette in cm.

Thermal unfolding and refolding curves were obtained by measuring the CD signal of 15 µM (≈0.1 mg/ml) protein buffered by either 10 mM K-phosphate (pH 7.0) or 10 mM K-acetate (pH 4.0), respectively, in a 1 cm quartz cuvette upon heating to 95°C and subsequently re-cooling to the initial temperature. The scan speed was 1°C/min, the dwell time 1 min, and the integration time 4 s. Checking the reversibility of thermal unfolding and determination of the wavelength used for

temperature transition curves was done by recording far-UV CD spectra at 25°C, then 95°C, and after subsequent re-cooling to 25°C in a 1 mm pathlength cuvette using 25 µM protein solutions in either 10 mM K-phosphate (pH 7.0) or 10 mM K-acetate (pH 4.0). The wavelength to follow a thermal transition corresponds to the wavelength >215 nm with the largest difference in the CD signal between folded and unfolded state and was chosen for each transition individually. Using wavelengths <215 nm led to noisy signals at high temperatures and resulted in non-interpretable data.

Changes in ellipticity ($\theta$) upon thermal unfolding were analyzed with a two-state model using Fit-o-Mat version 0.752 (*Möglich, 2018*) to obtain the melting temperature ($T_m$) and enthalpy change at $T_m$ ($\Delta H_u(T_m)$) of the transition (both fit parameters) (*Equation 2*):

$$\theta = f_N \cdot \left(y_N + m_N \cdot (T - T_m)\right) + (1 - f_N) \cdot \left(y_U + m_U \cdot (T - T_m)\right) \tag{2}$$

with $T$ being the absolute temperature in K, $y_N$ and $y_U$ the y-intercepts, and $m_N$ and $m_U$ the slopes of the N- and U-state baselines, respectively. $f_N$ is the fraction of folded molecules, which is related to the equilibrium constant $K_u$ according to *Equation 3*:

$$f_N = \frac{1}{1 + K_u} \tag{3}$$

Finally, $K_u$ is related to the change in Gibbs free energy of the unfolding reaction ($\Delta G_u$) and $\Delta H_u(T_m)$ by *Equation 4*:

$$K_u = e^{-\Delta G_u/(RT)} \text{ with } \Delta G_u = \Delta H_u(T_m) - \frac{T}{T_m} \cdot \Delta H_u(T_m) \tag{4}$$

where $R$ is the ideal gas constant.

CD-based chemical equilibrium unfolding experiments were performed at 25°C. Urea (BioScience Grade; ≈10 M) and GdmCl (≈8 M; both from Carl Roth GmbH & Co. KG, Karlsruhe, Germany) stock solutions were prepared according to *Pace et al., 1990*. Far-UV CD unfolding experiments were conducted using a 1 mm cuvette. All points of the unfolding curves were obtained from individual samples, each containing 40–60 µM (≈0.25–0.4 mg/ml) protein in either 10 mM K-phosphate (pH 7.0) or 10 mM K-acetate (pH 4.0), respectively. All samples were equilibrated over-night. The denaturant concentration of each sample was determined refractrometrically after CD data acquisition.

As for the thermal transitions, the wavelength to follow a chemical denaturation corresponds to the wavelength >215 nm with the largest difference in the CD signal between folded and unfolded state and was chosen for each transition individually (wavelengths <215 nm led to noisy signals and non-interpretable data at high denaturant concentrations).

Unfolding curves that indicate a two-state transition were analyzed using the linear extrapolation method (*Santoro and Bolen, 1988*) with Fit-o-Mat version 0.752 (*Möglich, 2018*) to obtain $\Delta G_u(H_2O)$ and the $m$ value (*Equation 5*):

$$S = f_N \cdot \left(y_N + m_N \cdot [\text{denat}]\right) + (1 - f_N) \cdot \left(y_U + m_U \cdot [\text{denat}]\right) \tag{5}$$

where $S$ is the signal derived from far-UV CD spectroscopy (i.e. the $\Theta_{MRW}$ value), intrinsic Trp fluorescence (for *Vc*RfaH-CTD), or the normalized peak volumes of the [$^1$H, $^{13}$C]-ctHSQC major/minor species signals for *Ec*RfaH-KOW residue S139, respectively. [denat] is the denaturant (i.e. urea or GdmCl) concentration in M, $y_N$ and $y_U$ are the y-intercepts, and $m_N$ and $m_U$, the slopes of the N- and U-state baselines, respectively. $f_N$ is given by *Equation 3*. In this case, $K_u$ is defined as (*Equation 6*):

$$K_u = e^{-\Delta G/(RT)} \text{ with } \Delta G = \Delta H(H_2O) - m \cdot [\text{denat}] \tag{6}$$

Finally, the [denat]$_{1/2}$ value is obtained by (*Equation 7*):

$$[\text{denat}]_{\frac{1}{2}} = \frac{\Delta G(H_2O)}{m} \tag{7}$$

Near-UV CD unfolding experiments of *Ec*RfaH-KOW were conducted using a 1 cm quartz cuvette and 0.5 mM protein in 10 mM K-phosphate (pH 7.0). As the exchange between folded and unfolded state is reasonably fast ($k_{ex} \approx 15$ s$^{-1}$ at 0 M urea/GdmCl), all points were obtained from a titration of the initial denaturant-free protein sample with a 10 M urea or 8 M GdmCl solution in 10 mM K-phosphate (pH 7.0).

The sample was then incubated for 5 min at 25°C to reach equilibrium. Curves were smoothed mathematically using a Savitzky-Golay filter.

To probe reversibility of chemical unfolding and validate incubation times used to reach equilibrium, proteins were dialyzed against 20 mM NH$_4$HCO$_3$ (pH 7.0) buffer, shock-frozen, lyophilized, and subsequently solved in 10 mM K-phosphate (pH 7.0) or 10 mM K-acetate (pH 4.0) with or without 10 M urea/8 M GdmCl, respectively. CD samples containing the identical denaturant concentration (1–2 samples in pre-transition region, 1 at [denat]$_{1/2}$, 1 in post-transition region) were then prepared from the native or unfolded proteins. All samples were equilibrated over-night; far-UV CD spectra were then recorded using a 1 mm quartz cuvette.

## Fluorescence spectroscopy

Fluorescence spectra were recorded at 25°C using a Peltier-controlled Fluorolog-3 fluorimeter (Horiba Europe GmbH, Oberursel, Germany) equipped with a 1 cm quartz cuvette (Hellma GmbH & Co. KG, Müllheim, Germany). Samples for chemical denaturation of *Vc*RfaH-KOW contained ≈11 μM protein and were prepared as described for the far-UV CD samples. The *Vc*RfaH-KOW Trp residue was excited at 295 nm; emission spectra were then recorded from 300 to 400 nm with slit widths between 2.65/2.65 and 2.8/2.8 nm (excitation/emission) and an integration time of 0.2 s. Analysis of the resulting denaturation curve was performed as described for CD data.

ANS (Sigma/Merck KGaA, Darmstadt, Germany) interaction experiments were conducted by preparing a urea denaturation series of *Ec*RfaH-KOW (final concentration: 5 μM) as described for the CD-based unfolding experiments, equilibrating over-night and adding ANS at a fluorophore:protein ratio of 100:1. Fluorescence spectra were then recorded from 410 to 650 nm following excitation at 395 nm with slit widths of 2.6/2.6 nm (excitation/emission) and 0.1 s integration time. A control experiment was conducted with identical experiment and instrument setup, respectively, but samples lacking protein. The obtained fluorescence at a given wavelength was then plotted against the urea concentration of the respective sample.

## Differential scanning calorimetry

The KOW domains were in either 10 mM K-acetate (pH 4.0; hSpt5-KOW5) or 10 mM K-phosphate (pH 7.0; all other domains), respectively. Given a lack of Trp residues in most domains, the protein concentration was determined via absorption at 205 nm using the molar extinction coefficient ($\varepsilon_{205}$) as calculated by the Protein Calculator tool (**Anthis and Clore, 2013**).

Initial DSC experiments were carried out on a MicroCal VP-DSC instrument (MicroCal/Malvern Panalytical, Malvern, UK; active volume: 509 μl). The samples were vacuum degassed at room temperature just before the measurements. Prior to the protein-buffer scans, several buffer-buffer scans were performed. All thermograms were recorded at a scan rate of 1.5 K/min under an excess pressure of 30 psi in passive feed-back mode from ≈10°C to 110°C or 130°C (*Mj*Spt5-KOW5), respectively. The unfolding was calometrically reversible for *Ec*NusG-KOW, *Mt*NusG-KOW, *Mj*Spt5-KOW, and *Ec*RfaH-KOW (data not shown). hSpt5-KOW5 aggregated at pH 7.0 upon unfolding at all tested concentrations, whereas *Vc*RfaH-KOW aggregated at concentrations >0.2 mg/ml.

We repeated the measurements for all proteins but *Mt*NusG-KOW using a MicroCal VP-Capillary DSC instrument (Malvern Panalytical, Malvern, UK; active volume 137 μl). The thermograms were obtained at a heating rate of 1.5 K/min with excess pressure (30 psi) and at mid gain feed-back mode. Buffer-buffer runs were done prior to the protein measurements. Thermograms were recorded from ≈5°C to 130°C. The protein concentration was 0.2–1 mg/ml for *Ec*NusG-KOW, 0.25–1 mg/ml for *Mj*Spt5-KOW, 0.15–0.25 mg/ml for hSpt5-KOW5, 0.2–1 mg/ml for *Ec*RfaH-KOW, and 0.1–0.15 mg/ml for *Vc*RfaH-KOW. The measurement for hSpt5-KOW5 was carried out with 10 mM K-acetate (pH 4.0), all other KOW domains were in 10 mM K-phosphate (pH 7.0).

The obtained raw DSC data (VP-DSC data for *Mt*NusG-KOW, VP-Capillary DSC data for all other KOW domains) was scan rate normalized, the corresponding buffer-buffer baseline was subtracted, and the thermograms were then normalized to 1 mol of protein. To extract the thermodynamic parameters, the data was fitted to a two-state unfolding model including a temperature-dependent change in heat capacity from native to unfolded state (**Viguera et al., 1994**). The temperature dependence of the native state heat capacity ($C_{p,0}$) is assumed to be linear (**Equation 8**; note that $C_{p,0}$ contains an instrument-specific offset), whereas the difference in heat capacity to the unfolded state ($\Delta C_{p,u}(T)$) is approximated by a parabolic function (**Equation 9**):

$$C_{p,0} = a_0 + b_0 \cdot T \tag{8}$$

$$\Delta C_{p,u}\left(T\right) = a + b \cdot T + c \cdot T^2 \tag{9}$$

The value for the pre-factor of the quadratic term, $c$, was obtained by calculating the theoretical partial molar heat capacity, $C_p(T)$, of the unfolded state for each of the six protein domains at 5°C, 25°C, 50°C, 75°C, 100°C, and 125°C, respectively, according to *Makhatadze and Privalov, 1990*. Then, the values for $C_p(T)$ were plotted over the temperature and a parabolic function was fitted, yielding $c$.

The concentration-normalized heat capacity ($C_p$) then is the sum of $C_{p,0}$, the change of the 'internal' heat capacity that depends on the fraction of the protein in the folded and unfolded state (i.e. the equilibrium constant $K_u$), $\delta C_p^{int}$, and the excess heat absorption of the unfolding reaction $\delta C_p^{exc}$ (*Equation 10*):

$$C_p = C_{p,0} + \delta C_p^{int} + \delta C_p^{exc} \tag{10}$$

With $\delta C_p^{int}$ and $\delta C_p^{exc}$ given in *Equation 11*:

$$\delta C_p^{int} = \Delta C_{p,u} \cdot \frac{K_u}{1+K_u} \text{ and } \delta C_p^{exc} = \frac{\left(\Delta H_u(T)\right)^2}{RT^2} \cdot \frac{K_u}{\left(1+K_u\right)^2} \tag{11}$$

$K_u$ is related to the change in Gibbs energy of the unfolding reaction ($\Delta G_u(T)$) by (*Equation 12*):

$$K_u = e^{-\Delta G_u(T)/(RT)} \text{ with } \Delta G_u\left(T\right) = \Delta H_u\left(T\right) - T \cdot \Delta S_u\left(T\right) \tag{12}$$

The temperature-dependent enthalpy and entropy change ($\Delta H_u(T)$, and $\Delta S_u(T)$, respectively) are given by *Equations 13 and 14*:

$$\Delta H_u\left(T\right) = \Delta H_u\left(T_m\right) + a \cdot \left(T - T_m\right) + \frac{b}{2} \cdot \left(T^2 - T_m^2\right) + \frac{c}{3} \cdot \left(T^3 - T_m^3\right) \tag{13}$$

$$\Delta S_u\left(T\right) = \frac{\Delta H_u(T_m)}{T_m} + a \cdot ln\frac{T}{T_m} + b \cdot \left(T - T_m\right) + \frac{c}{2} \cdot \left(T^2 - T_m^2\right) \tag{14}$$

During fitting of $C_p$, parameters $a_0$, $b_0$, $a$, $b$, $T_m$, and $\Delta H_u(T_m)$ were allowed to float, while $c$ was kept constant.

## Acknowledgements

We thank I Artsimovitch for providing pVS13 and F Werner for providing pGEX-2TK_*Mj*Spt5-KOW. Our sincere thanks go to FX Schmid for detailed discussions and valuable comments on the manuscript. We also thank A Matagne and Birgitta M Wöhrl for helpful discussions, CM Johnson for performing DSC measurements, A Häussermann for carrying out the NMR-based refolding experiment of *Vc*RfaH, and A Hager for technical support, as well as the Northern Bavarian NMR Centre (NBNC) for providing access to NMR spectrometers. The project was supported by the German Research Foundation grant Ro617/21-1 (to P Rösch) and COST action CA 15126 ARBRE-MOBIEU (to PKZ, TD, SHK).

## Additional information

### Funding

| Funder | Grant reference number | Author |
| --- | --- | --- |
| European Cooperation in Science and Technology | CA15126 | Philipp K Zuber |

| Funder | Grant reference number | Author |
|---|---|---|

The funders had no role in study design, data collection and interpretation, or the decision to submit the work for publication.

## Author contributions
Philipp K Zuber, Conceptualization, Formal analysis, Investigation, Visualization, Writing – original draft, Writing – review and editing, conceived the project; Tina Daviter, Investigation, Writing – original draft, Writing – review and editing, TD performed the DSC measurements and gave input to the manuscript; Ramona Heißmann, Investigation, cloned and produced the KOW domain constructs; Ulrike Persau, Investigation, cloned and produced the KOW domain constructs; Kristian Schweimer, Investigation, Writing – original draft, Writing – review and editing, performed and analyzed the NMR experiments and gave input to the manuscript; Stefan H Knauer, Conceptualization, Formal analysis, Investigation, Project administration, Supervision, Writing – original draft, Writing – review and editing, conceived and supervised the project

## Author ORCIDs
Philipp K Zuber (iD) http://orcid.org/0000-0001-5139-3930
Tina Daviter (iD) http://orcid.org/0000-0003-2636-5959
Kristian Schweimer (iD) http://orcid.org/0000-0002-3837-8442
Stefan H Knauer (iD) http://orcid.org/0000-0002-4143-0694

## Decision letter and Author response
Decision letter https://doi.org/10.7554/eLife.76630.sa1
Author response https://doi.org/10.7554/eLife.76630.sa2

# Additional files

## Supplementary files
- MDAR checklist
- Transparent reporting form

## Data availability
Coordinates for VcRfaH-KOW have been deposited to the Protein Databank (ID: 6TF4). Chemical shifts have been deposited in the Biological Magnetic Resonance Databank under the following accession numbers: #28039 (hSpt5-KOW5), #28040 (MjSpt5-KOW), #28041 (VcRfaH) and #34450 (VcRfaH-CTD). Source data files have been provided for Figures 2, 3, 5, and 6.

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
