## [Editor Report]

This fundamental and timely work provides insights into the structural basis and thermodynamics of fold-switching proteins involved in the antitermination of transcription. By comparing six fold-switching and single-folding KOW domains from different organisms the authors provide compelling evidence showing that fold-switching domains are less thermodynamically stable than their single-folding counterparts. Furthermore, the authors identify a second fold-switching member of the NusG superfamily, VcRfaH, and investigate the physical basis of this fold-switching transition. This work should be of great interest to scientists in the fields of protein folding (theory and experiment), structural biophysics, and advanced protein NMR spectroscopy.

---

## [Decision Letter]

**Decision letter after peer review:**

Thank you for submitting your article "Structural and thermodynamic analyses of the β-to-α transformation in RfaH reveal principles of fold-switching proteins" for consideration by *eLife*. Your article has been reviewed by 3 peer reviewers, one of whom is a member of our Board of Reviewing Editors, and the evaluation has been overseen by Volker Dötsch as the Senior Editor. The reviewers have opted to remain anonymous.

Essential Revisions:

1) The authors should provide direct experimental evidence for the presence of residual helical structure in VcRfaH-KOW, and a lack of such a structure in hSpt5-KOW5. The 15N chemical shift differences between the major and the minor states in VcRfaH-KOW and hSpt5-KOW are quite similar, and in both constructs the chemical shifts of the minor state deviate (at least to some degree) from random coil values. Thus, it is still possible that the minor species of hSpt5-KOW have helical propensity like Ec- and VcRfaH. Therefore, in order to strengthen the conclusion that VcRfaH-KOW is a fold-switching domain while hSpt5-KOW is a single folding domain (exchanging with an unfolded state), experimental measurements of secondary structure propensities should be obtained for both constructs. This can be achieved, for example, by recording 13Cα (and perhaps 13Cβ) CEST experiments for the VcRfaH-KOW and hSpt5-KOW5 constructs.

2) The secondary structure predictions for the different KOW domains appear to be somewhat inaccurate. For example, the prediction shows helical propensities for the single folding MtNusG-KOW domain, which are comparable to those found in fold-switching proteins EcRfaH-KOW and VcRfaH-KOW. Therefore, these predictions do not appear to provide strong support for the lack of fold-switching transitions in hSpt5-KOW5.

3) It is not clear why the authors state that the minor species of EcRfaH-KOW are in exchange between helical and completely unfolded conformations. The chemical shift differences in Figure 6 appear quite comparable, indicating one population. Furthermore, the presence of an ensemble of conformations in fast exchange on the NMR time scale would greatly complicate the analysis and interpretation of the chemical shift changes.

4) It is not clear why the CD transitions were analyzed at wavelengths of 220 – 230 nm, and not at a wavelength where folded and unfolded state show the biggest difference in ellipticity. 220-230 nm is typical for α-helical proteins, but the isolated KOW domains are all β-state under native conditions. Furthermore, the reasons for measuring the thermal denaturation (Figure 2, supplement 1) at different wavelengths should be explicitly stated.

*Reviewer #1 (Recommendations for the authors):*

Overall the paper is well done and the comparison between the different KOW domains, as well as their in-depth characterization are interesting. Also, while the paper focuses on the KOW domains, I am now quite interested to see how the isolated NGN domain would behave in the absence of KOW. Is this domain also marginally stable and does it undergo any conformational changes upon removal of the KOW?

The main thing that I find to be somewhat lacking is evidence for the presence of residual helical structure in VcRfaH-KOW, and a lack of such a structure in hSpt5-KOW5. As both of these constructs display a minor state in the CEST experiments with chemical shifts deviating (at least to some degree) from random coil values, I think direct evidence of residual secondary structure (or lack thereof) is quite necessary to support the main conclusion. This can be achieved by recording 13Cα (and perhaps 13Cβ) CEST experiments, not only for EcRfaH, but also for the VcRfaH-KOW and hSpt5-KOW5 constructs. This, in my opinion, would greatly add to the paper.

*Reviewer #2 (Recommendations for the authors):*

As mentioned under Strengths, the stability differences between EcNusG-KOW and EcRfaH-KOW are striking because they come from the same organism. The authors should consider highlighting this in their manuscript.

As mentioned under Weaknesses, the results indicate that VcRfaH-KOW and EcRfaH-KOW (fold-switching) and hSpt5-NusG (single-folding) have CEST intermediates with similar chemical shift differences. The authors should justify why differences in 15N chemical shifts > 2ppm indicate residual structure. As presented, it seems like an arbitrary cut-off. If the authors can justify this, it would help for them to put dashed lines at -2 and +2 ppm in Figure 6A and Figure 5 Supplement 1B and C. It doesn't look like very many chemical shifts exceed +/-2 for any of the 3 CTDs that show millisecond exchange from CEST. Based on this and the concerns stated in the public review, I would suggest that they focus their manuscript more on positive results (residual helical structure in the *E. coli* RfaH CTD) and leave the nature of VcRfaH-KOW and hSpt5-NusG-KOW for future work.

The presented secondary structure predictions are suspect since those of MtNusG-KOW (single-folding) are comparable to Ec- and VcRfaH (fold-switching). Thus, one cannot strongly argue that hSpt5-NusG has lower helical propensity from its secondary structure predictions. Based on this, the authors should either state that the hSpt5-NusG results may indicate that their proposed fold switching mechanism is incomplete or provide more experimental evidence for less helical structure in hSpt5-NusG.

The thermal denaturations were measured at different wavelengths (Figure 2, supplement 1). Sometimes the reason is obvious (no difference between folded and unfolded spectra at 222), other times it isn't. The authors should justify why they selected different wavelengths.

---

## [Author Response]

Essential Revisions:1) The authors should provide direct experimental evidence for the presence of residual helical structure in VcRfaH-KOW, and a lack of such a structure in hSpt5-KOW5. The 15N chemical shift differences between the major and the minor states in VcRfaH-KOW and hSpt5-KOW are quite similar, and in both constructs the chemical shifts of the minor state deviate (at least to some degree) from random coil values. Thus, it is still possible that the minor species of hSpt5-KOW have helical propensity like Ec- and VcRfaH. Therefore, in order to strengthen the conclusion that VcRfaH-KOW is a fold-switching domain while hSpt5-KOW is a single folding domain (exchanging with an unfolded state), experimental measurements of secondary structure propensities should be obtained for both constructs. This can be achieved, for example, by recording 13Cα (and perhaps 13Cβ) CEST experiments for the VcRfaH-KOW and hSpt5-KOW5 constructs.

We fully agree with the reviewers on the importance of these experiments. Thus, we made several attempts to determine the Ca chemical shifts of the *Vc*RfaH-KOW minor CEST species (new “Figure 6 —figure supplement 3”). However, we were able to detect a minor species dip only in a limited number of ^13^Cα-CEST traces of ^13^C,^15^N-*Vc*RfaH-KOW and those dips were quite small, despite a measuring time of more than a week, which is significantly higher than for the Cα-CEST of ^13^C-*Ec*RfaH-KOW (we added traces of the ^13^Cα-CEST of ^13^C,^15^N-*Ec*RfaH-KOW as new “Figure 6 —figure supplement 1”). We attribute this to the fact that the population of the minor species of *Vc*RfaH-KOW is much lower than the one of *Ec*RfaH-KOW (0.4 *vs.* 5.5 %). Due to limited sample stability, we were not able to increase the experimental time further. As fitting with ChemEx was not possible, we analyzed the CEST traces only qualitatively, i.e. we determined the chemical shifts of the minor species dips manually. Strikingly, the profiles that show a minor species dip belong to amino acids predominantly located in the region of amino acids #140-150, the region corresponding to region 2 in *Ec*RfaH-KOW and which exhibits high Δδ^15^N values. As the chemical shifts of the minor species dips are downfield-shifted as compared to random coil chemical shifts (new “Figure 6 —figure supplement 3”), these data suggest that also *Vc*RfaH-KOW has residual helical elements in this region. This is also supported by the secondary structure predictions (see #2 below). These results and conclusions were added to the manuscript.

We also carried out a Cα-CEST experiment on a ^13^C,^15^N-labeled sample of hSpt5-KOW5 (new “Figure 6 —figure supplement 4”). In this case, we observed dips that correspond to the minor species and analysis of the chemical shifts of these dips revealed that the minor species does not contain residual structure. As hSpt5-KOW is the only KOW domain in this study that is not located at the C-terminus of the protein we hypothesize that the decreased stability is a direct result of the lack of the physiological environment, which, in turn, is mirrored by the β-barrel being in equilibrium with an unfolded species. This data was included in the manuscript and the Results section on the minor species of hSpt5-KOW5 was modified accordingly.

2) The secondary structure predictions for the different KOW domains appear to be somewhat inaccurate. For example, the prediction shows helical propensities for the single folding MtNusG-KOW domain, which are comparable to those found in fold-switching proteins EcRfaH-KOW and VcRfaH-KOW. Therefore, these predictions do not appear to provide strong support for the lack of fold-switching transitions in hSpt5-KOW5.

We agree that the secondary structure predictions by Net-CSSP were not unambiguous as the algorithm predicted also helical propensity for some regions of *Mt*NusG-KOW. Thus, we performed another secondary structure prediction by JPred 4 and added the results, giving new Figure 6 —figure supplement 2. JPred 4 clearly predicted 4-5 β-strands for all NusG/Spt5-KOW domains with high confidence whereas multiple secondary structure propensities were predicted for the RfaH-KOW domains. As we provide experimental evidence for the absence of helical structures in the minor CEST species of hSpt5-KOW5 (see reply to “Essential Revisions #1”), we now base our argumentation that hSPt5-KOW5 is not a fold-switching protein on the experimental findings rather than on the secondary structure predictions. We changed this section in the manuscript accordingly.

However, the difference in the secondary structure predictions clearly demonstrates two things: first, secondary structure predictions strongly depend on the algorithm. Second, ambiguous secondary structure predictions are a necessary, but not a sufficient condition for a protein to switch folds. We included this conclusion in the Discussion section on the requirements for fold-switching proteins.

3) It is not clear why the authors state that the minor species of EcRfaH-KOW are in exchange between helical and completely unfolded conformations. The chemical shift differences in Figure 6 appear quite comparable, indicating one population. Furthermore, the presence of an ensemble of conformations in fast exchange on the NMR time scale would greatly complicate the analysis and interpretation of the chemical shift changes.

We agree that the chemical shift differences between the minor species signals resulting from CEST experiments and the unfolded state are similar to the chemical shift differences of the minor species derived from the chemical denaturation experiments and the unfolded state (for both ^15^N and ^13^C). Thus, the minor species detected in the CEST experiments corresponds to the weak signals in the HSQC spectra at 0 M urea. Due to the presence of two dips in the CEST profiles the traces can be well described via a two-state model, indicating that the minor species can be described as one population/macrostate at the CEST/HSQC timescale. However, just as for CPMG experiments the increased ^15^N *R*_2_ values imply further exchange processes within the minor species on the µs-ms time scale. Thus, we interpret the minor species as an ensemble of interconverting, largely unfolded species with helical propensity in regions 1 and 2. Additionally, the NMR-based denaturation experiments show that the addition of urea shifts the minor species signals linearly towards the positions of a completely unfolded species. Of course, we do not know if the minor species is already in a dynamic equilibrium with a completely unfolded species in the absence of urea (as mentioned, we observe just two dips in the CEST traces, but maybe the population of the completely unfolded state is just below the detection limit or the exchange rate is not on the CEST time scale). Based on the increased ^15^N *R*_2_ values in regions 1 and 2 and the linear transition towards the unfolded state in the urea denaturation series we hypothesize, however, that the minor species is an ensemble composed of largely unstructured states with one extremum being a completely unfolded state (referred to as state “U”) and the other one being an unstructured species exhibiting helices in regions 1 and 2 (referred to as a-helical unfolding intermediate Uα). We are perfectly aware that this interpretation is somewhat speculative, but we find it plausible and it can very nicely explain all effects observed during the chemical denaturation. In order to make clear which conclusion is experimentally evidenced and which interpretation is more speculative we rewrote the corresponding section.

4) It is not clear why the CD transitions were analyzed at wavelengths of 220 – 230 nm, and not at a wavelength where folded and unfolded state show the biggest difference in ellipticity. 220-230 nm is typical for α-helical proteins, but the isolated KOW domains are all β-state under native conditions. Furthermore, the reasons for measuring the thermal denaturation (Figure 2, supplement 1) at different wavelengths should be explicitly stated.

The wavelength was chosen with the aim to optimize the signal that was used to follow the denaturations, especially given the fact that all KOW domains are exclusively β proteins. The largest signal difference between the folded and the unfolded state was mostly around 200 nm (as can be seen, for example, for the thermal denaturations in Figure 2 – Supplement 1). Thus, we initially followed all denaturations at 200 nm (data not included). However, the signal became extremely noisy at high temperatures/ high denaturant concentrations and data were not interpretable. Thus, we chose the wavelength on a case-by-case basis in order to optimize the signal, i.e. we measured at/near the wavelength of the highest difference in the CD signal above 215 nm, although we are aware that it is not ideal to vary the wavelength. We included the information on the choice of the wavelength in the Material and Method section as well as the legend of Figure 2, Figure 2 —figure supplement 1 and Figure 3.

Reviewer #1 (Recommendations for the authors):Overall the paper is well done and the comparison between the different KOW domains, as well as their in-depth characterization are interesting. Also, while the paper focuses on the KOW domains, I am now quite interested to see how the isolated NGN domain would behave in the absence of KOW. Is this domain also marginally stable and does it undergo any conformational changes upon removal of the KOW?

Indeed, it would be highly interesting to see how the NGN domain behaves in the absence of KOW. We have been addressing this issue for some time, but, unfortunately, the isolated NGN domain of *Ec*RfaH has very limited stability and solubility (< 2 µM), most probably due to the highly hydrophobic KOW/β’ clamp helices binding surface. Nevertheless, in future, we plan to study the isolated domain by NMR spectroscopy using more sensitive experiments and specific labeling techniques (e.g. methyl group labeling) and addressing *Vc*RfaH-NGN, which might behave differently.

The main thing that I find to be somewhat lacking is evidence for the presence of residual helical structure in VcRfaH-KOW, and a lack of such a structure in hSpt5-KOW5. As both of these constructs display a minor state in the CEST experiments with chemical shifts deviating (at least to some degree) from random coil values, I think direct evidence of residual secondary structure (or lack thereof) is quite necessary to support the main conclusion. This can be achieved by recording 13Cα (and perhaps 13Cβ) CEST experiments, not only for EcRfaH, but also for the VcRfaH-KOW and hSpt5-KOW5 constructs. This, in my opinion, would greatly add to the paper.

We fully agree with the reviewer’s comment and addressed the suggestion as described in our reply to “Essential Revisions #1”.

Reviewer #2 (Recommendations for the authors):As mentioned under Strengths, the stability differences between EcNusG-KOW and EcRfaH-KOW are striking because they come from the same organism. The authors should consider highlighting this in their manuscript.

We added a paragraph to the Discussion section “requirements for fold-switching proteins” highlighting this particularity.

As mentioned under Weaknesses, the results indicate that VcRfaH-KOW and EcRfaH-KOW (fold-switching) and hSpt5-NusG (single-folding) have CEST intermediates with similar chemical shift differences. The authors should justify why differences in 15N chemical shifts > 2ppm indicate residual structure. As presented, it seems like an arbitrary cut-off. If the authors can justify this, it would help for them to put dashed lines at -2 and +2 ppm in Figure 6A and Figure 5 Supplement 1B and C. It doesn't look like very many chemical shifts exceed +/-2 for any of the 3 CTDs that show millisecond exchange from CEST. Based on this and the concerns stated in the public review, I would suggest that they focus their manuscript more on positive results (residual helical structure in the *E. coli* RfaH CTD) and leave the nature of VcRfaH-KOW and hSpt5-NusG-KOW for future work.

We agree with the concerns of the reviewer. The threshold of 2 ppm to indicate residual structure was chosen arbitrarily. We removed the argumentation based on this cut-off. Instead, we carried out Ca-CEST experiments with ^13^C,^15^N-*Vc*RfaH-KOW and ^13^C,^15^N-hSpt5-KOW5 and included our findings as described in our reply to “Essential Revisions #1”.

The presented secondary structure predictions are suspect since those of MtNusG-KOW (single-folding) are comparable to Ec- and VcRfaH (fold-switching). Thus, one cannot strongly argue that hSpt5-NusG has lower helical propensity from its secondary structure predictions. Based on this, the authors should either state that the hSpt5-NusG results may indicate that their proposed fold switching mechanism is incomplete or provide more experimental evidence for less helical structure in hSpt5-NusG.

We agree with the reviewer that the presented secondary structure predictions are not a strong basis to argue that the minor species of hSpt5-KOW5 has no helical structure. Thus, we added another secondary structure prediction to show differences in those predictions (see reply to “Essential Revisions #2) as well as experimental evidence for the absence of helical structures in the hSpt5-KOW5 minor species (see reply to “Essential Revisions #1).

The thermal denaturations were measured at different wavelengths (Figure 2, supplement 1). Sometimes the reason is obvious (no difference between folded and unfolded spectra at 222), other times it isn't. The authors should justify why they selected different wavelengths.

We chose the wavelength to monitor thermal and chemical denaturations based on the maximum signal difference between folded and unfolded state at a wavelength > 215 nm as described in our reply to “Essential Revisions #4”